**Production and accumulation of reef framework by calcifying corals and macroalgae on**
**a remote Indian Ocean cay.**
M. James McLaughlin[1], Cindy Bessey[1], Gary A. Kendrick[2], John Keesing[1,2], Ylva S.
Olsen[1,2],
[1] CSIRO Oceans & Atmosphere, Indian Ocean Marine Research Centre, 64 Fairway,
Crawley 6009 Australia
[2] School of Biological Sciences and The Oceans Institute, University of Western Australia,
Indian Ocean Marine Research Centre, 64 Fairway, Crawley 6009 Australia
Correspondence to: james.mclaughlin@csiro.au

**Abstract**

Coral reefs face increasing pressures in response to unprecedented rates of environmental change at present. The coral reef physical framework is formed through the production of calcium carbonate ($CaCO_3$) and maintained by marine organisms, primarily hermatypic corals, and calcifying algae. The northern part of Western Australia, known as the Kimberley, has largely escaped land-based anthropogenic impacts and this study provides important metabolic data on reef-building organisms from an undisturbed set of marine habitats. From the reef platform of Browse Island, located on the mid-shelf just inside the 200 m isobath off the Kimberley coast, specimens of the dominant coral (6 species) and algal (5 species) taxa were collected and incubated ex-situ in light and dark shipboard experimental mesocosms for 4 hours to measure rates of calcification and production patterns of oxygen. During experimental light/dark incubations, all algae were net autotrophic producing 6 to 111 mmol $O_2$ m$^{-2}$ day$^{-1}$. In contrast, most corals were net consumers of $O_2$ with average net fluxes ranging from $-42$ to 47 mmol $O_2$ m$^{-2}$ day$^{-1}$. The net change in pH was generally negative for corals and calcifying algae ($-0.01$ to $-0.08$ h$^{-1}$). Resulting net calcification rates (1.9 to 9.9 g $CaCO_3$ m$^{-2}$ d$^{-1}$) for corals, and calcifying algae (*Halimeda* and *Galaxura*) were all positive and were strongly correlated to net $O_2$ production. In intertidal habitats around Browse Island, estimated relative contributions of coral and *Halimeda* to the reef production of $CaCO_3$ were similar at around 600 to 840 g m$^{-2}$ year$^{-1}$. The low reef platform had very low coral cover of $< 3\%$ which made a smaller contribution to calcification of ~240 g $CaCO_3$ m$^{-2}$ year$^{-1}$. Calcification on the subtidal reef slope was predominantly from corals, producing ~1540 g $CaCO_3$ m$^{-2}$ year$^{-1}$, twice that of *Halimeda*. These data provide the first measures of community metabolism from the offshore reef systems of the Kimberley. The relative contributions of the main reef builders, in these undisturbed areas, to net community metabolism and $CaCO_3$ production is important to understand exclusively climate-driven negative effects on tropical reefs.

## 1. Introduction

Coral reefs in the Anthropocene era have been degraded for more than a century by overfishing and pollution, but now even remote reefs (where local pressures are low) face increasing stresses through anthropogenic climate change (Hughes et al., 2017b). With the currently unprecedented rate of environmental change, coral reefs face growing pressures in response to eutrophication (Hewitt et al., 2016), recurrent large scale weather events (marine heat waves, etc.), sedimentation (Hughes et al, 2017a), and rising atmospheric greenhouse gases (especially carbon dioxide, $CO_2$; IPCC, 2014) that result in increasing ocean temperatures (due to atmospheric heat absorption) and ocean acidification (OA) (Hoegh-Guldberg, 2007; Doney et al., 2009; Perry et al., 2018). The pressures of global climate change are causing shifts in the composition of coral reef species, and the urgent focus now is on identifying, quantifying and maintaining reef ecosystem function so that coral reefs can continue to persist and deliver ecosystem services into the future (Harborne et al., 2017).

The functioning of healthy coral reefs, as some of the world's most biologically (Stuart-Smith et al., 2018) and structurally complex ecosystems (Hughes et al., 2017b), results in a number of ecosystem services. They provide coastal protection, with reef structures acting to dampen wind and wave driven surges (Perry et al., 2018). Reefs support a diverse range of species that provide critically important resources, such as food, for coastal livelihoods (Hoegh-Guldberg et al., 2007). As one of the most important determinants of overall reef function, the construction and maintenance of the calcium carbonate ($CaCO_3$) reef structure (the accumulation of which requires the net production of calcium carbonate by resident taxa; Cornwall et al., 2021), is vital to the myriad of ecosystem services that coral reefs provide (Hoegh-Guldberg et al., 2007; Andersson et al., 2013; Moberg and Folke, 1999).

Community metabolism on a reef is a combination of the photosynthesis and dark respiration
of the organisms that live there. Coral reefs are known for their high calcification and
photosynthetic production, and measurements of reef metabolism make it possible to
characterize reef health in terms of these fundamental processes. These functions are dependent
on the maintenance of the framework structure of the reefs. Photosynthesis fixes $CO_2$ in organic
materials, whereas the reverse reaction, dark respiration, releases it. Overall, the excess organic
production in a coral reef community (i.e., the difference between gross primary production
and dark respiration) acts as a $CO_2$ sink, while calcification acts as a source of $CO_2$ (Lewis,
1977; Kinsey, 1985). Despite the drawdown of $CO_2$ during the day via photosynthetic
processes, most reef flats are sources of $CO_2$ to the atmosphere due to their low net fixation of
$CO_2$ and rather large release of $CO_2$ by precipitation of calcium carbonate (Ware et al, 1992;
Gattuso et al, 1993; Gattuso et al, 1995; Smith, 1995; Frankignoulle et al, 1996; Gattuso et al,
1996b). One notable exception to this is in algal-dominated reef communities, which are sinks
for atmospheric $CO_2$. They exhibit larger excess community production and/or a lower
community calcification, (e.g., Kayanne et al, 1995; Gattuso et al, 1996a; Gattuso et al, 1997).
Photosynthesis and calcification both consume inorganic carbon, but a proportion of $CO_2$
generated by calcification can be used for photosynthetic carbon fixation, so the combined
processes can be viewed as reciprocally supportive (Gattuso et al., 1999).

The coral reef physical framework is formed through the production of calcium carbonate
($CaCO_3$) and maintained by marine organisms, primarily hermatypic corals, crustose coralline
algae (CCA), and other calcifying algae (Vecsei, 2004; Perry et al., 2008; Perry et al., 2012).
Scleractinian corals are primary reef builders in tropical environments, producing $CaCO_3$
through skeletal deposition. This net calcium carbonate production is a balance between gross
production minus the loss due to physical, chemical, and biological erosion (Cornwall et al.,
2021). The net calcium carbonate production and related potential vertical accretion of reefs is
increasingly threatened by anthropogenic climate change (Perry et al., 2018). For scleractinian
corals, one of the most significant consequences of OA is the decrease in the concentration of
carbonate ions ($CO_2^{-3}$) (Kleypas and Yates, 2009). Coral skeletons are made from the mineral
phase of calcium carbonate (aragonite), and the saturation state of aragonite ($\Omega_{arg}$) is often
related to rates of calcification. Studies have demonstrated that, as $CO_2$ concentrations rise, the
saturation state of aragonite ($\Omega_{arg}$) decreases and, in turn, the rate at which corals calcify
declines (Schneider and Erez, 2006; Langdon, 2005; Pandolfi et al., 2011; Venn et al., 2013).
Projections suggest that future rates of coral reef community dissolution may exceed rates of
$CaCO_3$ production (calcification), leading to net loss (Silverman et al., 2009; Hoegh-Guldberg
et al., 2007) with the majority of coral reefs unable to maintain positive net carbonate
production globally by 2100 (Cornwall et al., 2021).

In scleractinian corals with zooxanthellae, the precipitation of $CaCO_3$ through calcification is
tightly coupled to photosynthetic fixation of $CO_2$ and on average tends to be three times
higher in daylight conditions than in darkness (Gattuso et al., 1999). Calcification rates can
increase further through feeding on phytoplankton and suspended particles (Houlbreque and
Ferrier-Pages, 2009). Change in community structure is linked to the balance between
community metabolism and calcification with the $CO_2$ flux of seawater (Kayanne et al.,
2005). In reefs under thermal stress, rates of primary production and dark respiration
increase, but community excess organic production decreases dramatically (Kayanne et al.,

109  2005).


Reef algae are also an often-overlooked important structural component of coral reef
ecosystems. Their morphological diversity provides food (Overholtzer and Motta, 1999),

habitat and shelter (Price et al., 2011) for a number of invertebrate and fish species, with productivity sustaining higher trophic levels. Reef-building corals are generally considered to be the dominant components of healthy or pristine coral reefs, but inconspicuous turfing and encrusting coralline algae contribute substantially to reef benthic primary resources in these areas (Odum and Odum, 1955; Hatcher, 1997). The abundance of large frondose macroalgae is typically inversely related to coral abundance (Done, 1992; Hughes et al., 2017b); macroalgae are common on reef flat, back reef, and inshore fringing reef areas, whereas corals are more common on reef slopes (Purcell and Bellwood, 2001). Calcified macroalgae can also contribute significantly to the deposition of carbonates (Nelson, 2009). In particular, species of the genus *Halimeda* (order Bryopsidales), widely distributed across tropical and subtropical environments, contribute significantly to reef calcification and productivity rates because of their fast growth and rapid turnover rates (Vroom et al., 2003, Smith et al., 2004, Nelson, 2009) compared to corals or coralline red algal (CRA). Calcification rates of *Halimeda* make it a major contributor to $CaCO_3$ in reefs in the Caribbean (Blair and Norris, 1988; Nelson, 2009), Tahiti and the Great Barrier Reef (Drew, 1983; Payri, 1988). In certain locations, precipitation of calcium carbonate can approach 2.9 kg $CaCO_3$ $m^{-2}$ $yr^{-1}$, positioning *Halimeda* as a major contributor to carbonate budgets within shallow waters around the globe (Price et al., 2011). This group further occupies a diverse range of environments (mangroves, seagrass beds, and coral reefs) and can produce structurally complex mounds that serve as critical habitat for a diversity of marine life (Rees et al., 2007).

The corals and algae dominating the benthos of these complex ecosystems have the potential to change the local chemistry of the water column (Duarte et al., 2013), superseding larger scale oceanographic and atmospheric influences (Kleypas et al., 2011). Metabolic processes can deplete or replenish oxygen, carbon, and nutrient concentrations either within

hydrodynamic boundary layers over time (Shashar et al., 1993; Zeebe et al. 1999; Anthony et
al., 2011; Shamberger et al., 2011) or in larger water masses as they move across a given reef
(Barnes, 1983; Barnes and Lazar, 1993; Frankignoulle et al., 1996; Gattuso et al., 1996a;
Niggel et al., 2010; Wild et al., 2010). The magnitude of reef contributions to changes in water
column chemistry is difficult to predict because of the net effect of local oceanographic
conditions, relative abundance of the different members of the reef community and their
individual metabolic rates. In addition to producing and consuming $O_2$, photosynthetic
organisms alter concentrations of dissolved inorganic carbon through uptake of dissolved
inorganic carbon ($CO_2$ or bicarbonate ion; e.g. Raven et al., 1995) during photosynthesis and
release of $CO_2$ during dark respiration, thus altering the pH of the surrounding water column
(Murru and Sandgren, 2004). Calcifying organisms also alter the biogeochemistry in the water
column by releasing $CO_2$ and $H^+$ ions during the production of $CaCO_3$ and thus decreasing the
pH (Jokiel, 2011). The effect on water column chemistry by hybrid organisms like calcifying
primary producers, such as corals with zooxanthellae and calcifying algae, therefore becomes
very challenging to measure in situ.

Coastal environments are frequently high-use areas by humans, impacted by multiple land- and
sea-based human activities, and in such cases the potential for interaction between climate and
other anthropogenic variables affecting biological responses exists (Harley et al., 2006;
Schindler, 2006; Walther, 2010). Contrary to Southwestern Australia which has one of the
fastest increasing rates of change from cumulative human impacts (Halpern et al., 2019), the
Kimberley bioregion located in the northern part of Western Australia is unique, representing
one of the few "very low impact" tropical coast and shelf areas globally – only 3.7% of the
global oceans fall in this category (Halpern et al., 2008). It is host to extensive coastal reef
systems, isolated offshore reefs and islands. Few process studies have been carried out in the
region due to the remoteness of these reefs, some of which are located 100s of km from the
coastline, meaning that fieldwork and data acquisition can be difficult and costly. So that reefs
can continue to deliver ecosystem services into the future metabolic measurements of reef
organisms are necessary to characterize reef health in terms of fundamental processes such as
photosynthesis, respiration and calcification (Madin et al., 2016; Carlot et al., 2022). However,
there are limited numbers of studies examining the individual effects of key primary producers
on water chemistry in the same study, and thus, we lack knowledge of the relative contributions
of the main reef builders to net community metabolism and $CaCO_3$ production on most coral
reefs. Here, we compare metabolic and calcification rates of the dominant intertidal taxa of
macroalgae and coral at Browse Island, a small island in the Kimberley, something never
previously examined in these systems. Rates of metabolism and calcification were determined
in on-ship incubations in October 2016, April 2017 and October 2017. Using the proportional
cover of the dominant benthic community, these rates were upscaled to gain whole of
community metabolism estimates for the island habitats.

**2. Methods**
*2.1 Study site*
Browse Island is located on the mid-shelf just inside the 200 m isobath off the Kimberley coast
in northern Western Australia (14°6'S, 123°32'E; Fig. 1). The island is surrounded by a small
(~ 4.5 $km^2$) planar platform reef consisting of a shallow lagoon, an extensive reef flat that is
conspicuously absent to the northeast of the island, and a well-defined reef crest and slope.
Tides are semidiurnal with a maximum range of < 5 m, exposing the reef crest and reef platform
habitats during low tides. The intertidal habitats are characterised by low species richness and
dominated by small turfing algae and calcified macroalgae of the genus *Halimeda* (15–22%
and 6–9% cover respectively) (Olsen et al., 2017). Coral assemblages are well developed with
cover of 5–8% in the intertidal habitats and 18% on the shallow reef slope (< 10 m) (Olsen et
al., 2017).

*2.2 Algae and coral collection*
Specimens of the dominant coral and algal taxa were collected from the reef platform by hand
during low tide, immediately brought back to the vessel and kept in a holding tank with
circulating seawater. Macroalgae included the calcifying green alga *Halimeda opuntia*, which
was the dominant species of *Halimeda* on the reef platform, the green alga *Caulerpa* sp., and
the calcifying red alga *Galaxaura* sp. Pieces of turf algae (turf) as well as turf attached to a
piece of rock (turf + substrate) were measured. In April 2016, drift algae of the genus
*Sargassum* found floating on the water surface were also included although this taxa was not
been found growing anywhere on the reef. Hermatypic corals included *Pocillopora* sp.,
*Goniastrea* sp., *Porites* sp., *Heliopora* sp., *Acropora* sp. and *Seriatopora* sp. Whole pieces of
coral small enough to fit inside the incubation cores (inner diameter ~90 mm) were collected
to minimise tissue damage. All coral samples were > 50 mm diameter and therefore
operationally defined as adults and estimated to be at least 2 to 7 years old depending on the
taxa (Trapon et al., 2013).

*2.3 Light and dark incubations*
Light and dark incubations were undertaken on the back deck of the research vessel. Four 60
L holding tanks were placed in a shade-free spot under natural light conditions, filled with
seawater and connected to a flow-through seawater system driven by an Ozito PSDW-350 watt
Dirty Water Submersible Water Pump with a maximum flow rate of 7,000 litres/hour, which
ensured the setup remained at ambient temperature (Fig. 2). The intensity of photosynthetically
active radiation (PAR) was recorded for each set of incubations with a HOBO Micro Station
logger (H21-002, Onset) placed inside one of the tanks. Six 1.56 L clear Perspex incubation
cores (24 total per incubation) fitted with stirring caps, were placed in each holding tank and
spaced evenly apart to minimise shading (Fig. 2).

Depending upon abundance, individual specimens of algae and coral were placed in 6 to 12
replicate incubation cores per taxa except where not enough individuals could be found. Table
1 shows the taxa incubated during each sampling trip and the number of replicates. Water
samples from the holding tanks were measured at each time point as controls and, in addition,
in October 2017, a separate seawater control (six replicate incubation cores with seawater) was
included. After a period of acclimation (1 to2 h), incubations were run over a four-hour period.
The light incubations were conducted while the sun was at its zenith providing full irradiance
to the samples. After two hours, the tubs were covered with a black lid ensuring no light could
enter and the samples incubated for two hours in the dark.

To estimate oxygen production or consumption during the incubations, a 40 mL water sample
was extracted from each of the 24 cores and the four tubs at the start of the incubations and
hourly thereafter. A port in the cap of each core allowed for sample collection using a syringe.
As the sample was removed, the same volume of liquid was automatically replaced from the
flowthrough tank into the core so that the core volume remained constant through the
experiment. Samples were immediately analysed for temperature and dissolved oxygen ($O_2$)
with a YSI 5100 bench-top oxygen and temperature meter with YSI 5010 BOD stirring probe,
calibrated daily in air. Sample pH was determined using a TPS Aqua pH meter with an Ionode
probe, calibrated daily with pH 7.00 and 10.00 buffers. A second 35 mL water sample was
collected from each core and tub and split between one 10 mL glass vacutainer for alkalinity
and duplicate 10 mL sterile vials for nutrient analyses. Nutrient samples were immediately
frozen and alkalinity samples were stored cool and dark. At the end of the incubation, algal and
coral specimens were frozen. All samples were transported to Perth, Western Australia, to be
analysed.

*2.4 Surface areas of coral and algal specimens*
Metabolic measurements were standardised by surface area of the incubated specimens since
this represents the area available for photosynthesis and nutrient uptake. The surface area of
specimens of coral, *Halimeda* and turf + substrate were estimated using a single wax dipping
method (Veal et al., 2010). Specimens were dried, weighed and then dipped in paraffin wax at
65°C. The waxed samples were weighed again, and the weight of the wax calculated. The
surface area was estimated from the wax weights against a calibration curve constructed by
wax dipping geometric wooden objects of known size. The surface areas of the remaining taxa,
were estimated from photographs in ImageJ (Rueden et al., 2017). The 'footprint' of each
sample, i.e. the surface area of reef occupied by the organism, was also estimated by tracing
the outline of the specimen photographed from straight above in ImageJ.

*2.5 Chemical analyses*
Concentrations of nitrate + nitrite (hereafter referred to as nitrate), ammonium, phosphate and
dissolved silica in water samples were analysed in duplicate by flow injection analysis (Lachat
QuickChem 8000) with detection by absorbance at specific wavelengths for silica [QuikChem
Method 31-114-27-1-D], nitrate [Quikchem Method 31-107-04-1-A] and phosphate
[QuikChem Method 31-115-01-1-G]), and by fluorescence for ammonia according to Watson
et al. 2005. Detection limits were 0.02 $\mu$mol $L^{-1}$ for all inorganic nutrient species, with a
standard error of $< 0.7\%$.

From SOP3b in Dickson et al. 2007, total alkalinity was determined for single replicates to the
nearest 5 μmol $L^{-1}$ equivalent (hereafter referred to as μmol $L^{-1}$) using an open cell Metrohm
titrator (841 Titrando, Burette: 800 Dosino 10 mL) with a Metrohm micro-glass pH probe
calibrated with Certipur buffer solutions at pH 2.00, 4.01, 7.00, and 10.00 (at 25.0°C). Samples
were kept in a Jubalo F12 temperature control water bath prior to decanting a 10 mL aliquot of
sample into a vessel with a water jacket maintaining temperature at 25.0°C. Samples were
titrated with 0.012 N HCl, standardised against sodium carbonate (99.95 to100.05 wt%) with
an initial volume of titrant added to reach pH 3.5. Titrations were run to an end-point of pH 3
with Gran plot (Excel macro) to determine the total alkalinity endpoint near pH 4.2. Carbonate
system parameters were calculated from pH (measured during the incubations) and total
alkalinity using the package 'seacarb' (Gattuso et al., 2018) in R (R Core Team, 2018).
Alkalinity and carbonate parameters were not determined in April 2016.

*2.6 Oxygen fluxes and calcification rate calculations*
The changes in $O_2$ concentrations during light- and dark incubations were expressed as mmol
per day assuming stable hourly production rates over 24 h. Any replicates where $O_2$ did not
increase during both of the light intervals or did not decrease during both of the dark intervals
were excluded from further analysis. Net fluxes of $O_2$ per day (mmol $day^{-1} m^{-2}$) were calculated
for each sample assuming a 12 h photoperiod. Calcification rates of corals and calcifying algae
(*Halimeda opuntia*. and *Galaxaura* sp.) were estimated using the alkalinity anomaly method
(Smith and Key, 1975) uncorrected for changes in nutrient concentration (Chisholm and
Gattuso, 1991) where precipitation of one mole of $CaCO_3$ leads to the reduction of total
alkalinity by two molar equivalents. Rates per surface area (mmol $day^{-1} m^{-2}$) were obtained by
dividing these values by the surface area of each specimen.

A census-based approach was used to estimate the amount of $CaCO_3$ and $O_2$ produced by a
single taxon per unit area of reef surface per year (Shaw et al., 2016). The rates of calcification
and net $O_2$ production per day were divided by the 'footprint' area of each specimen. To
estimate the relative contributions from each taxon to community production per $m^2$ of reef,
these rates were multiplied by the relative percent cover in each of the major habitats. Estimates
of percent cover based on drop camera image analysis were obtained from Olsen et al. (2017).
The productivity rates for individual coral species were combined into one value for coral.

*2.7 Statistical analyses*
The relationships between net changes in pH and $O_2$ and between net $O_2$ production and net
calcification (in light and dark incubations) were examined by linear regression. Significance
of regressions were calculated for algae, calcified algae and corals and the 95% confidence
intervals for the slope of each line in R (R Core Team, 2018). Regressions were examined with
ANOVA and deemed significant if $p < 0.05$.

**3   Results**
*3.1 Experimental conditions*
Nutrient concentrations were low and similar among sampling trips (Table 2), as is
characteristic of tropical Eastern Indian Ocean offshore waters (McLaughlin et al., 2019).,
Concentrations of nitrate were 0.05 to 0.17 $\mu$mol $L^{-1}$, ammonium 0.12 to 0.13 $\mu$mol $L^{-1}$,
phosphate 0.07to 0.1 $\mu$mol $L^{-1}$, and silicate 2.3 to 3 $\mu$mol $L^{-1}$. Oxygen was around 0.19 mmol
$L^{-1}$ to 0.22 mmol $L^{-1}$ and salinity 34.2 to34.8 ppt. Light and temperature conditions in the
incubations were representative of *in situ* conditions on the reef platform and were similar
among trips. PAR levels were 1500 to 1587 $\mu$E $m^{-2}$ $s^{-1}$ and slightly higher in October.
Temperatures were 28.3 to 32.8°C and highest in April. Carbonate system parameters were
not obtained for April 2016 due to instrument error, and some minor differences in pCO2,
HCO3$^-$, CO3$_2$$^-$, DIC and Ω Aragonite were noted between October 2016 and 2017 (Table 2).
Alkalinity and pH were both higher in 2016, and there were associated minor differences in
the concentrations of the carbonate species and the aragonite saturation state (Table 2).

*3.2 Changes in oxygen and pH*
Changes in dissolved $O_2$ differed among taxa, and between light and dark incubations. In the
seawater controls $O_2$ changed by $< 0.01$ mmol h$^{-1}$ in both light and dark incubations, showing
that the contribution of any organisms in the seawater itself to $O_2$ production and dark
respiration was minimal. No corrections were therefore applied. In the light incubations $O_2$
productivity fluxes were positive for all taxa (Fig. 3, top panel). The highest light flux of $O_2$
of ~380 mmol m$^{-2}$ day$^{-1}$ was measured for *Galaxaura* in October 2017 (Fig. 3, top). Corals
generally produced 100 to 260 mmol $O_2$ m$^{-2}$ day$^{-1}$ in the light, except *Heliopora,* which had
a flux of 50 to 80 mmol $O_2$ m$^{-2}$ day$^{-1}$. All taxa consumed $O_2$ during the dark incubations
when changes in $O_2$ are due to dark respiration, with mean fluxes of $-15$ to $-190$ mmol $O_2$
m$^{-2}$ day$^{-1}$ (Fig. 3, middle). All algae were net autotrophic and produced 6 to 111 mmol $O_2$
m$^{-2}$ day$^{-1}$ with the highest net $O_2$ flux measured for *Galaxaura* and turf at 111 and 36 mmol
$O_2$ m$^{-2}$ day$^{-1}$ respectively (Fig. 3, bottom). In contrast, around half of the corals were net
consumers of $O_2$ and average net fluxes spanned a wide range from $-42$ to 47 mmol $O_2$ m$^{-2}$
day$^{-1}$.

In the light incubations, pH generally increased by 0.03 to 0.25 h$^{-1}$ for all taxa, except for
*Halimeda* in April 2016 and October 2017, which showed no change or a very small increase
(Fig. 4, top panel). In dark incubations, mean pH decreased for all taxa by 0.02 to 0.21 h$^{-1}$
indicative of a net increase in $CO_2$ through dark respiration (Fig. 4, middle). Non-calcifying
algae (*Sargassum*, *Caulerpa* and turf) raised net pH by 0.02 to 0.05 $h^{-1}$ (assuming equal
periods of light and darkness) (Fig. 4, bottom panel). The net change in pH was generally
negative for corals and calcifying algae ($-0.01$ to $-0.08$ $h^{-1}$), except for the coral *Goniastrea*
in April and October 2016 (0.01 $h^{-1}$) and the calcifying alga *Galaxaura* (0.03 $h^{-1}$; Fig. 3,
bottom).

Net changes in pH are largely driven by metabolic uptake and release of $CO_2$. We found
positive relationships between changes in pH and net production or consumption of $O_2$ except
in seawater controls where changes in $O_2$ and pH were minor (Fig. 5). The relationships for
algae, calcifying algae and coral were all significant, but had relatively low adjusted $r^2$ values
of 0.59, 0.46 and 0.19 respectively, suggesting significant variability among species and
individuals within each of these groups.

*3.3 Calcification Rates*
Corals, *Halimeda* and *Galaxaura* had positive calcification rates in light ranging from 4.2 to
18.4 g $CaCO_3$ $m^{-2}$ $d^{-1}$ (Fig. 6, top panel). In the dark, calcifying rates were smaller and just
under half of the rates were negative suggesting dissolution of $CaCO_3$ (Fig. 6, middle panel).
The resulting net calcification rates (based on equal periods of light and dark_- monthly
average sunrise and sunset at Browse Island of 0552 and 1739 for April, and 0519 and 1754
for October; WillyWeather, 2022) were all positive and ranged from 1.9 to 9.9 g $CaCO_3$ $m^{-2}$
$d^{-1}$ (Fig. 6, bottom). Rates of calcification were strongly linearly correlated to net $O_2$
production and were significantly higher in light than in darkness for both corals and algae
(Fig. 7).

*3.4 Contributions to community production*
In intertidal habitats (lagoon and high reef platform) around Browse Island, the estimated
relative contributions of coral (8 % cover) and *Halimeda* (7 % cover) to the reef production
of $CaCO_3$ were similar, around 600 to 840 g $m^{-2}$ $year^{-1}$ (Fig. 8, top panel). The low reef
platform had very low coral cover of < 3% (Fig. 8, middle), which therefore made a smaller
contribution to calcification of ~240 g $CaCO_3$ $m^{-2}$ $year^{-1}$ in this habitat (Fig. 8, top). In
contrast, calcification on the subtidal reef slope was predominantly from corals (19 % cover),
which produced ~1540 g $CaCO_3$ $m^{-2}$ $year^{-1}$, around twice the amount compared to *Halimeda*
(7 % cover). *Galaxaura*, which had high measured rates of productivity and calcification, was
extremely rare (0.02 % total cover found only in October 2017; Olsen et al., 2017) and thus
its contribution to community calcification and productivity were negligible. Turf was
responsible for the majority of the $O_2$ production in all habitats and produced an estimated 8
to 13 mmol $O_2$ $m^{-2}$ $d^{-1}$ compared to < 2 for *Halimeda* mmol $O_2$ $m^{-2}$ $d^{-1}$ and −4 to −1 mmol $O_2$
$m^{-2}$ $d^{-1}$ for corals (Fig. 8, second panel from top).

4    **Discussion**
This study investigated the metabolism of coral and algae on the reef of remote Browse Island,
found on the mid-shelf region of the Kimberley in Western Australia. Due to its remoteness,
Browse Island presented a unique opportunity to observe these organisms in a pristine habitat
where direct anthropogenic pressures are minimal. The Island has semidiurnal tides reaching
a maximum range of 5 m (Olsen et al., 2017), half the magnitude of tides experienced by reefs
closer to the coast (McLaughlin et al., 2019), and its benthic structure is very different from
both Kimberley inner and outer shelf reefs. Lowe et al. (2015) have revealed that strongly tide-
dominated circulation can occur on Kimberley reef platforms and the trapping of water on a
reef, such as that found at Browse Island, can provide benefits for reef organisms in terms of
avoiding aerial exposure. However, it can dramatically increase the residence (or flushing)
times of reefs, which can lead to extreme diel variations in water quality (Lowe et al., 2015).
Seawater $O_2$ and carbonate chemistry can vary over diel tidal cycles, like those found at Browse
Island, and are related to patterns in autotrophic photosynthesis and dark respiration (e.g.,
Duarte et al., 2013). Primary production and the uptake of $CO_2$ by coral and algae during
daylight hours results in elevated pH and an elevated aragonite saturation state ($\Omega_{arag}$) during
the day when calcification rates peak. The process of calcification decreases pH in the
surrounding water, but for calcifying autotrophs $CO_2$ uptake and fixation through
photosynthesis can potentially offset changes to the carbonate chemistry caused by
calcification (Anthony et al., 2011; Smith et al., 2013).

Mesocosm experiments have shown that reef-building (hermatypic) corals tend to reduce pH
and consume $O_2$ (e.g. (Gattuso *et al.* 2015; Smith *et al.* 2013)), whereas calcifying macroalgae
increase pH and $O_2$ during daytime (Borowitzka and Larkum 1987; Smith *et al.* 2013). Both
corals and calcifying macroalgae reduce pH and $O_2$ concentrations due to respiration during
nighttime, but the rates of change differ among species (Smith *et al.* 2013). The organisms
investigated in the present study showed typical patterns of $O_2$ production in daylight and
consumption in darkness to other similar island reef systems as a result of photosynthesis and
dark respiration, but the metabolic measurements showed clear differences among taxonomic
groups. Algae had higher positive net $O_2$ fluxes with rates of 18 to 350 $\mu$mol $O_2$ m$^{-2}$ day$^{-1}$, of
which the red calcifying alga *Galaxaura* sp. had the highest rate of net productivity by far. For
corals, the relatively high $O_2$ increase measured in daylight was coupled with high rates of
respiration in darkness, creating a negligible or negative net $O_2$ production for most species,
except *Porites* sp. in April 2016 and *Seriatopora* sp. in October 2016 and 2017 which were net
positive. Although autotrophic, our data indicates that the majority of the corals we studied
utilise heterotrophic supply through feeding to help sustain growth in addition to
photosynthesis by zooxanthellae (Houlbreque and Ferrier-Pages, 2009). These patterns are
generally in agreement with those reported elsewhere, for example, fleshy and calcifying algae
showed net diel $O_2$ production, whereas corals generally consumed $O_2$ , i.e. were net
heterotrophic, on islands in the South Pacific (*Porites* sp.) and the Caribbean (*Madracis* sp.)
(Smith et al., 2013).

Concurrent with changes in $O_2$ were changes in seawater pH, where pH increased in daylight
(except for *Halimeda* in April 2016 where no change was measured) and decreased in darkness.
The effects of metabolic activity on bulk pH (uptake and release of $CO_2$ through photosynthesis
and dark respiration) cannot be directly separated from that of calcification, which is associated
with the release of $H^+$ ions thereby decreasing pH (Jokiel, 2011). However, differences were
observed in the net pH change in incubations between calcifiers and non-calcifiers. The net
effect of non-calcifiers on seawater pH was positive while the majority of calcifiers caused net
pH to decline. In the present study, *Halimeda* (April 2016) and *Goniastrea* (April and October
2016) caused relatively minor increases in pH, whereas the calcifying alga *Galaxaura* elevated
pH by, on average, 0.03 units, comparable to the net effect of non-calcifiers. This is not
surprising given the high rate of $O_2$ production measured for *Galaxaura*, which is associated
with sufficient levels of $CO_2$ fixation to compensate for the reduction in pH associated with
calcification in this species. A strong link was observed between metabolism and pH in all taxa,
demonstrated as linear relationships between changes in pH and $O_2$ during the incubations.
Previous research by Smith et al. (2013) identified two broad patterns: metabolic changes in
$O_2$ in non-calcifiers (fleshy and turf algae) linked to large changes in pH (steep slopes), and
metabolic changes in $O_2$ in calcifying organisms (*Porites* sp. *Madracis* sp. and *Halimeda* sp.)
producing little or no change in pH (shallow slopes). This is contrary to the present study's
observations where pH and $O_2$ relationship gradients were similar for calcifiers and non-
calcifiers. Non-calcifying organisms were found to consistently have a net positive effect on
both pH and $O_2$. Change in pH for the same net change in $O_2$ was elevated for non-calcifiers
compared to calcifiers.

Production and accumulation of reef framework carbonate is controlled by the relative rates of,
and the interactions between, a range of ecologically, physically and chemically driven
production and erosion processes (Perry et al., 2008; Montaggioni and Braithwaite, 2009), with
the relative importance of different taxa for $CaCO_3$ production differing among reefs and
among habitats within reefs. Coral growth can be measured in several ways: linear extension
rate, global skeletal growth and calcification rate (measured using the alkalinity technique or
by $^{45}Ca$ incorporation) (Houlbreque and Ferrier-Pages, 2009). Methods to calculate
calcification can vary in accuracy where overestimates of calcification rates can result from
calculations based on changes in alkalinity, while those relying on $CaCO_3$ content and growth
measurements, either through staining or tagging segments, may produce minimum estimates
as loss of new tissue is not accounted for (Hart and Kench, 2007; Houlbreque and Ferrier-
Pages, 2009). The alkalinity method employed in the present study was the best possible option
when working in a remote location where actual growth rates cannot be easily assessed, or use
of radioisotopes limited. Rates of net community calcification for reef flats worldwide range
from 7.3 to 90 mol (730 to 9000 g) $CaCO_3$ $m^{-2}$ $year^{-1}$ with an average of 47 mol (4700 g)
$CaCO_3$ $m^{-2}$ $year^{-1}$ (Atkinson, 2011). The patterns found in the present study — higher
calcification rates in daylight compared to in darkness for all corals and calcifying algae — are
typical. However, the coral $CaCO_3$ production rates per reef area (7 to 8% cover low reef
platform, 19% reef slope) measured here (240 g $m^{-2}$ $year^{-1}$ for low reef platform, 610 to 756 g
$m^{-2}$ $year^{-1}$ in the other intertidal habitats, and 1536 g $m^{-2}$ $year^{-1}$ on the reef slope) were
somewhat lower than values reported elsewhere. In 2016, the dark rates of calcification in
corals were less than 50% of the rates in light with some (*Porites* and *Heliopora)* negative.
Dark rates of calcification in 2017 were negative or near zero for all species except *Porites,*
*Pocillopora* and *Seriatopora*. Houlbreque et al. (2004) showed that coral feeding enhances dark
calcification rates in scleractinian corals, but incubations in our study were done in absence of
supplemental feeding. The trend observed here may be due to some dissolution of $CaCO_3$ due
to the reduced pH during dark incubations or could be an artefact of the experimental
conditions. This result should therefore be taken with some caution, in particular for *Porites* in
October 2016, which saw the largest decrease (Fig. 5, middle panel). However, the resulting
strong relationship between net carbonate production and net carbonate consumption is
consistent with previous studies both *in situ* and in mesocosms (Albright et al., 2013).

Corals are typically the primary framework-producing components on a tropical reef and
dominate carbonate production per unit area (Vecsei, 2004), however additional $CaCO_3$ is
produced by calcareous crustose coralline algae (CCA) and calcareous algae of the genus
*Halimeda*, (e.g. Payri, 1988). Sprawling lithophytic species of *Halimeda*, like the majority of
the *Halimeda* around Browse Island, tend to be fast growing and have high calcification rates
(Hart and Kench, 2007). Rates of calcification per area of 100% *Halimeda* cover have been
estimated to 400 to 1667 g $CaCO_3$ $m^{-2}$ $year^{-1}$ (in Hart and Kench, 2007 Suppl info). In other
locations, *Halimeda* has been estimated to contribute around 1100 to 2400 g $CaCO_3$ $m^{-2}$ $year^{-1}$
to benthic carbonate production (Drew, 1983; Freile et al., 1995; Hudson, 1985; Kangwe et al.,
2012; Payri, 1988; Rees et al., 2007), which is higher than the 600 to 840 g $CaCO_3$ $m^{-2}$ $year^{-1}$
estimated for *Halimeda opuntia* in the intertidal habitats in the present study. These rates
depend both on the intrinsic calcification rates and on the abundance or cover of algae (6.1 to
8.7% cover on Browse, which corresponds to ~150 to 250 g dw $m^{-2}$).

Nutrient capacity is one important driver of productivity in many reef ecosystems. The rate at
which nutrients are recycled between the constituents of the system (the ambient nutrient
availability, and the nutrients stored within plant and animal biomass) depends on input from
a variety of sources (e.g., associated with seasonal rains or upwelling) (DeAngelis, 1992;
Hatcher, 1990). Coral reefs, typically have low ambient nutrient availability and receive little
sustained exogenous nutrient input (Hatcher, 1990; Szmant, 2002), thus the high rates of
production found within these ecosystems are largely attributed to the nutrients stored and
cycled by living biomass (Pomeroy, 1974; DeAngelis et al., 1989; Sorokin, 1995). Fishes
typically make up a substantial component of living biomass on coral reefs and represent an
important reservoir of nutrients in these ecosystems (Allgeier et al., 2014). Contrary to our
expectations given its remote location in an area of apparently low anthropogenic impacts, the
reef platform around Browse Island was depauperate with a conspicuous lack of diversity in
key groups including macroalgae, macroinvertebrates and teleost browsers (Bessey et al.,
2020). McLaughlin et al. (2019) found surface water standing stock nutrient concentrations
low along Kimberley shelf. Conditions at Browse Island were similar with low water column
nutrients for nitrate, ammonia and phosphate during all trips. Understanding how changes in
animal populations alter nutrient dynamics on large ecological scales is a relatively recent
endeavour (Doughty et al., 2015). Allgeier et al. (2016) showed that targeted fishing of higher
trophic levels reduces the capacity of coral reef fish communities to store and recycle nutrients
by nearly half. Fish-mediated nutrients enhance coral growth (Meyer et al., 1983) and primary
production (Allgeier et al., 2013), and may regulate nutrient ratios at the ecosystem scale
(Allgeier et al., 2014).

The Kimberley region-wide averages of coral cover and macroalgal cover are 23.8% and 7.1%
(Richards et al., 2015) respectively. However, this relationship at Browse Island is reversed,
with macroalgae more dominant at 28% total cover to that of coral at 9% total cover. On the
Browse Island reef platform, the same pattern is observed where averages were 5 to 8% for
coral and 32% for macroalgae, differing from those of the regional averages of 14.4% and
15.5% of coral and macroalgae respectively (Richards et al., 2015). While the estimates
provided here approximate the relative contributions of *Halimeda* and coral to $CaCO_3$
production, they do not add up to a whole system budget. There are other organisms likely to
contribute significantly. For example, the present study did not measure metabolic or
calcification rates of encrusting coralline algae, which, although making up a modest 1.0 to
3.0% of the benthic cover in the lagoon and reef platform habitats at Browse Island, become
more prominent at 11.8 to 14.1% on the reef crest and slope (Olsen, unpublished data). To
calculate the true $CaCO_3$ production per area of reef, the calcification rate would need to be
multiplied by the benthic cover of coralline algae and the square of the benthic rugosity (Eakin,
1996). Using typical values for rugosity from Eakin (1996) of 1 to 1.4 for the lagoon and reef
platform and 1.7–2 for the reef crest and slope, and assuming a typical calcification rate of
1500 to 2500 g $m^{-2}$ $year^{-1}$ (for 100% flat-surface cover) (Hart and Kench, 2007), the
contribution of encrusting coralline algae to calcification in the lagoon and reef platform would
be minor at 70 to 134 g $CaCO_3$ $m^{-2}$ $year^{-1}$. However, they could produce a significant amount
of 980 to 1360 g $CaCO_3$ $m^{-2}$ $year^{-1}$ on the reef crest and slope, which is somewhere in between
the production rates estimated for *Halimeda* and corals. Encrusting coralline algae may
therefore contribute significantly to the $CaCO_3$ budget at Browse Island, at least in deeper
habitats. These values are similar to those measured elsewhere, for example 870 to 3770 g
$CaCO_3$ $m^{-2}$ $year^{-1}$ at Uva reef in the eastern Pacific (Eakin, 1996).

Metabolic rates of primary producers are clearly influenced by a multitude of factors including
hydrodynamics, irradiance, and nutrient availability (Smith et al., 2013). We were able to detect
considerable diurnal changes in water chemistry due to metabolic rates, since our experiments
were conducted in small enclosed mesocosms. The effect of metabolism on water chemistry is
expected to dissipate downstream in a more turbulent or dynamic environment (Anthony et al.
2011). However, coral and algae metabolic rates and resultant flux from diffusive boundary
layer also increases with flow rates (Carpenter et al. 1991; Lesser et al. 1994; Bruno and
Edmunds 1998; Mass et al. 2010). Because our experiments were conducted in near no-flow
chambers (mesocosm water was replenished with fresh seawater in small amounts during
sample extraction), our measurements are conservative values and likely represent the lower
range of potential effects that these reef organisms have on surrounding water chemistry,
however where residence times can be extended, particularly when trapping of water on the
reef at low tides occurs, our results are likely reflective of how these benthic organisms affect
water chemistry in the lagoonal habitats of Browse Island.

**5   Conclusions**
Browse Island is the only emergent mid-shelf reef in the Kimberley bioregion and is host to a
different benthic community composition compared to the closest reefs both inshore (e.g.
Montgomery Reef, Adele and Cassini Islands) and offshore (e.g. Ashmore Reef and Rowley
Shoals). The relative contributions of algae and corals to reef productivity are likely to differ
across the shelf, with corals becoming more important in offshore waters and algal calcifiers
being important on the mid-shelf. Estimated aerial production rates did not take into account
the relief (differences in height from place to place on the reef surface) of the substrate. The
reef platform surrounding Browse Island has relatively low surface relief, whereas the reef
slope and crest have high rugosity, which means production rates in the latter environments
may be underestimated. Despite these limitations, the rates estimated in this study are similar
to those measured elsewhere.

The higher cover of *Halimeda* and the low coral cover at Browse Island compared to other
reefs in the region mean that corals and *Halimeda* contribute equally to productivity rates of
$CaCO_3$ on the Browse Island reef flat, however, their relative contributions to the reef
framework and sedimentary budget of the reef is unknown. To gain an understanding of the
relationships between carbonate production and sinks on the reef, further study into the types
and amounts of $CaCO_3$ material found in each reef sink is necessary. The Kimberley coastal
shelf, which is characterised by coral reef environments with clear, low nutrient waters and
low productivity, has largely escaped land-based anthropogenic impacts, but has been
negatively affected by climate-driven coral bleaching and mortality, for example from heat
waves at Scott Reef in 1998 and 2016 (Smith et al., 2008, Gilmour et al., 2013 and Hughes et
al., 2017) and Ashmore Reef in 2003 and 2010 (Ceccarelli et al., 2011 and Heyward, 2011).

There is lack of sufficient observations of pCO2, nutrients and research on the upper ocean
carbon cycle from the Indian Ocean (Sreeush et al., 2020), and which are critical to modelling
of ocean acidification in the region (Panchang and Ambokar, 2021). The uptake of carbon
dioxide by the ocean alters the composition of seawater chemistry with elevated partial
pressures of carbon dioxide (pCO2) causing seawater pH and the $CaCO_3$ saturation state to
decrease (Feely et al, 2004). Ocean acidification directly threatens crucial trophic levels of
the marine ecosystem. Baseline reef measurements in undisturbed areas like Browse Island
are important to understand exclusively climate-driven stressors in lieu of local
anthropogenic pressures normally associated with coastal tropical reefs. The effects of
temperature stressors on reef communities and their productivity remain to be investigated in
this region. The effects of temperature stressors on reef communities and their productivity
remain to be investigated in this region. Different components of the reef around Browse
Island are likely to have different vulnerabilities to warming and heat waves. Future
environmental stressors leading to changes in benthic community composition, structure and
subsequent changes in reef productivity and in rates of production of $CaCO_3$, could have
major implications for Browse Island.

**Author contribution:** M. James McLaughlin – Conceptualization, formal analysis,
investigation, resources, methodology, visualisation, and writing (original draft preparation);
Cindy Bessey - Investigation, resources, project administration, and writing (review and
editing); Gary A. Kendrick - Conceptualization, funding acquisition, project administration,
supervision, and writing (review and editing); John Keesing - Conceptualization, funding
acquisition, investigation, resources, supervision, and writing (review and editing); Ylva S.
Olsen - Conceptualization, formal analysis, investigation, project administration, resources,
methodology, visualisation, and writing (original draft preparation)

**Declaration of funding:** The authors acknowledge the financial support of Shell Australia
Pty Ltd and the INPEX-operated Ichthys liquefied natural gas (LNG) project in conducting
this research.

**Competing interests:** The authors declare that they have no conflict of interest.


**Acknowledgements**
The authors thank Max Rees, Mark Tonks for their support of this work, the staff at Quest
Maritime for vessel logistics and the crews of the Browse Invincible and the Browse Express
for help in the field.
**Figures**

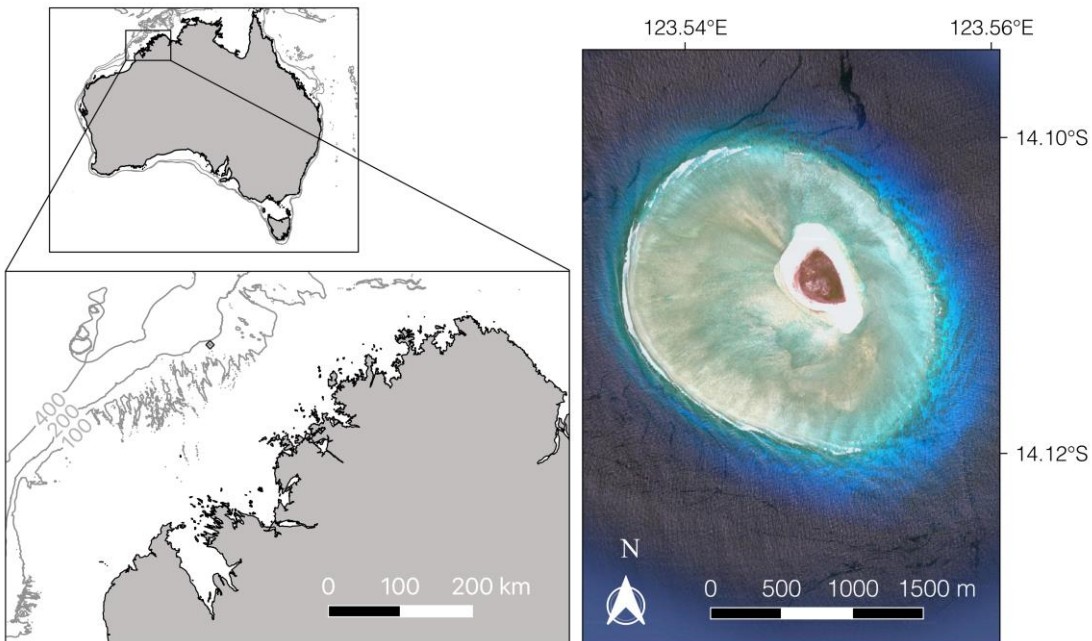


**Figure 1.** The study site, Browse Island (diamond, bottom left map), is located just inside the
200-m isobath on the continental shelf. The small map (top left) shows the location of the island
relative to the Australian coastline with the 100, 200 and 400 m isobaths marked in gray. The
satellite image (right; © Google Earth 2018) shows the extent of the reef.


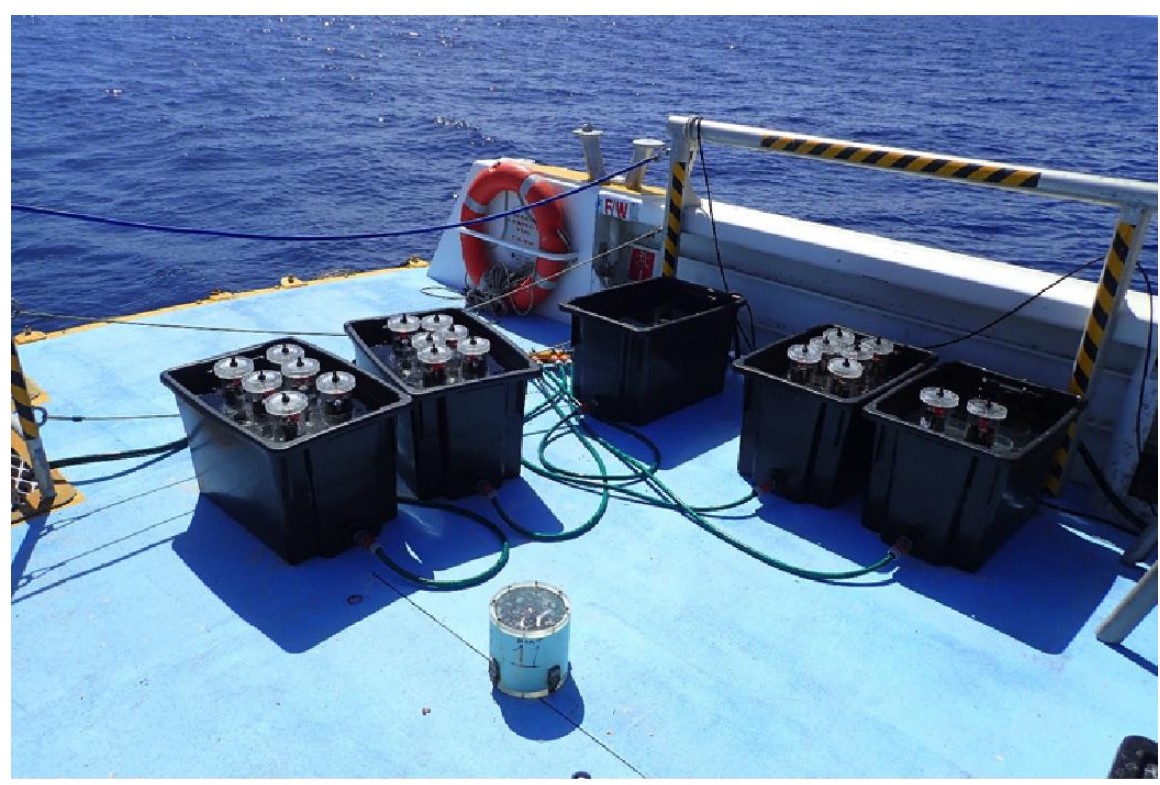

**Figure 2.** Experimental setup of respirometry incubations for Browse Island coral and macroalgae.


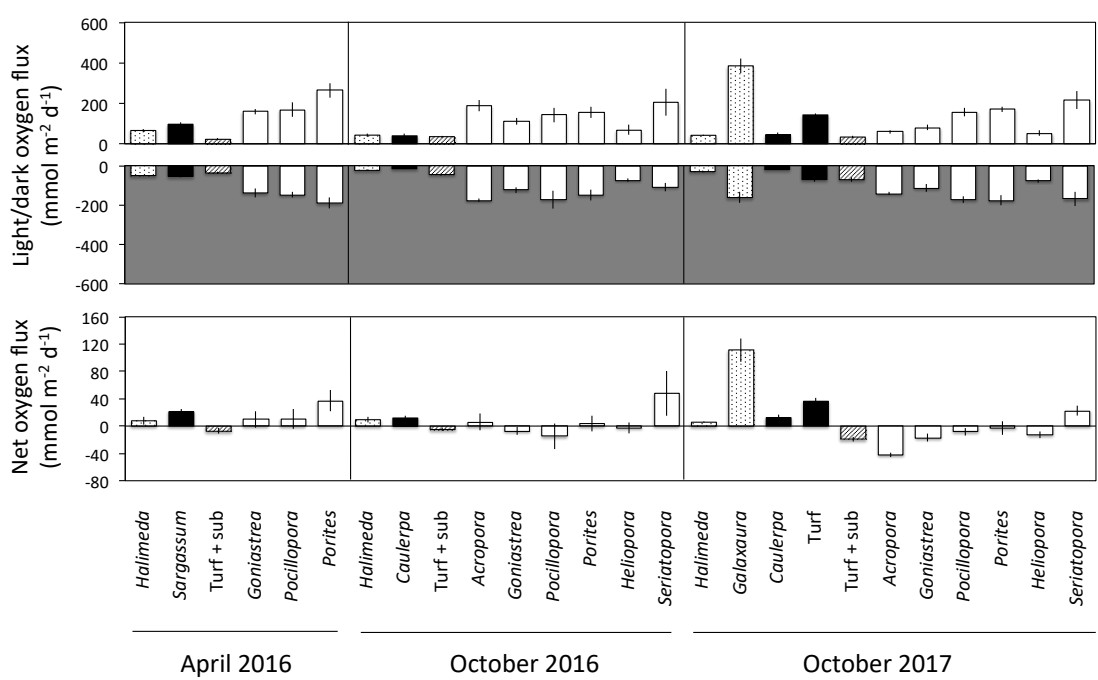


**Figure 3.** Net changes in oxygen (means ± se) in light (top) and dark (middle) incubations of calcifying algae (stippled), macroalgae and turf (black), turf + substrate (diagonal stripes) and coral (white) standardised by specimen surface area. The bottom panel shows the net daily production of oxygen (means ± se) assuming a 12-h photoperiod and stable rates of photosynthesis and dark respiration over a 24-h period.


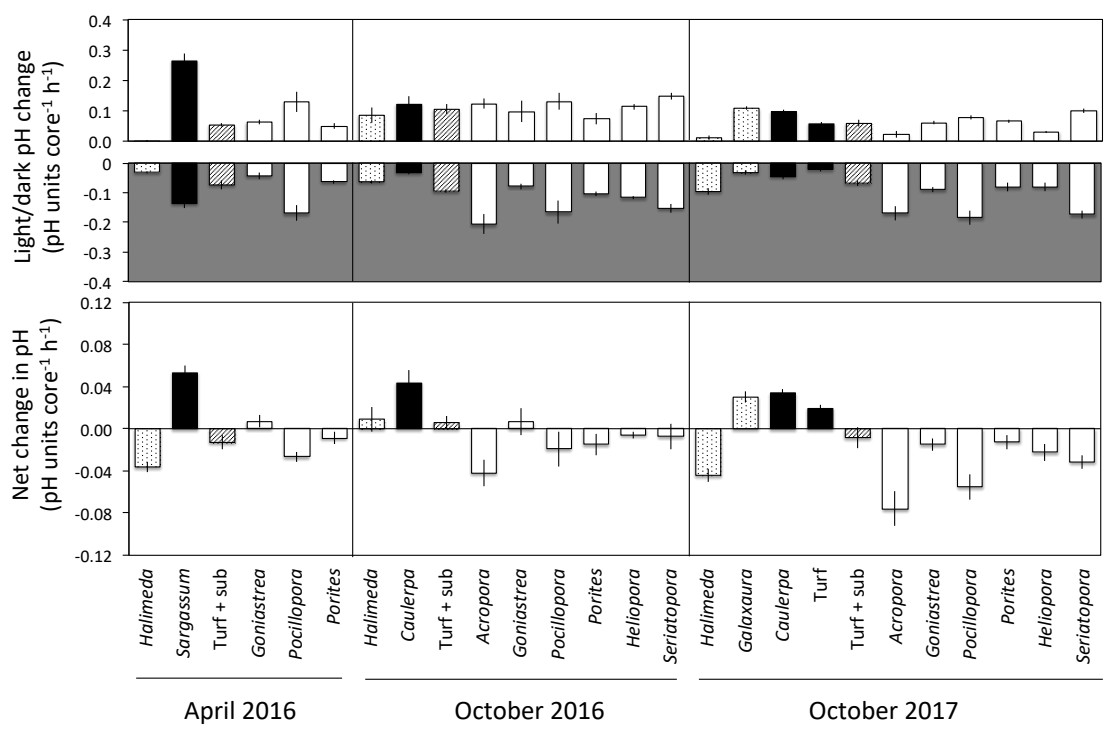


**Figure 4.** Net changes in pH per hour for each 1.56-L incubation core (means ± se) in light
(top) and dark (middle) incubations calcifying algae (stippled), macroalgae and turf (black),
turf + substrate (diagonal stripes) and coral (white). The bottom panel shows the net change in
pH per hour (means ± se) assuming equal periods of light and darkness.

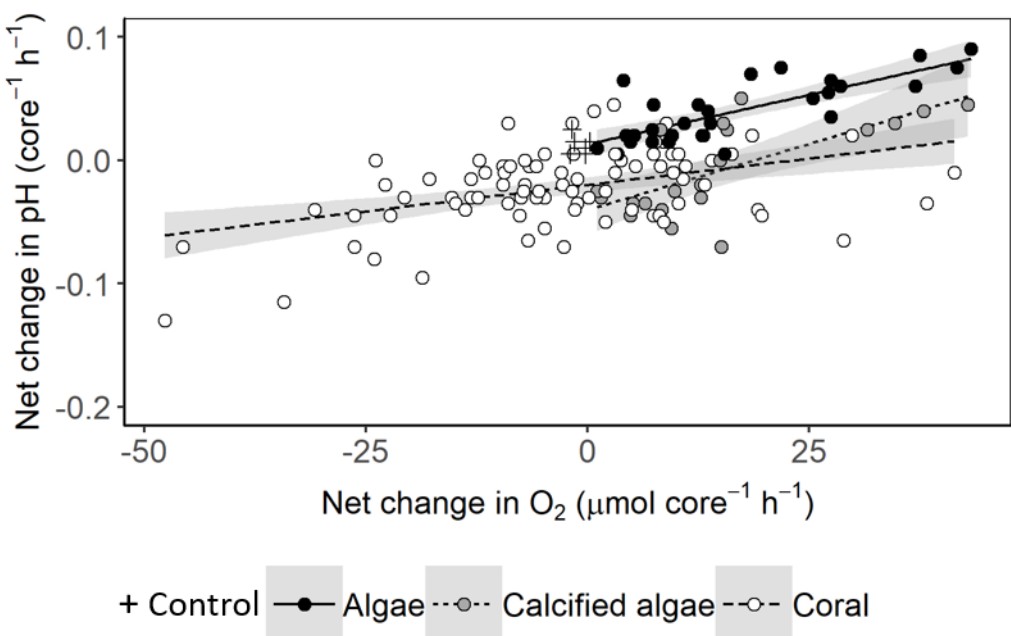


**Figure 5.** Net change in pH versus $O_2$ per 1.56-L incubation core assuming equal periods of
light and darkness. Linear relationships are fitted with 95% confidence intervals shown in gray.
For algae; net change in pH = 0.13 + 0.0016 × net change in $O_2$ (ANOVA: $F_{1,27}$ = 41.15, p
<0.001). For calcified algae; net change in pH = −0.04 + 0.0021 × net change in $O_2$ (ANOVA:
$F_{1,19}$ = 17.86, p <0.001). For corals; net change in pH = −0.02 + 0.00086 × net change in $O_2$
(ANOVA: $F_{1,82}$ = 18.88, p <0.001).

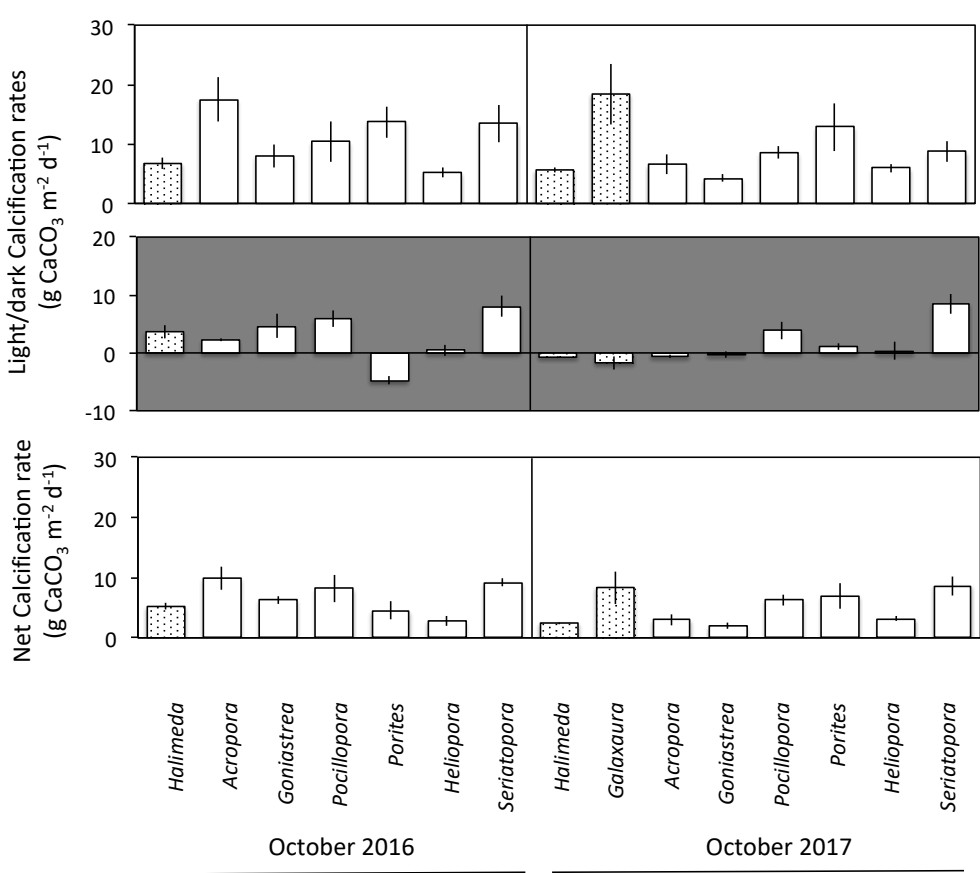


**Figure 6.** Calcification rates for corals (white) and calcifying algae (stippled) (means ± se) in

light (top) and dark (middle). The bottom panel shows the daily net calcification rate (means ±

se) assuming a 12-h photoperiod.


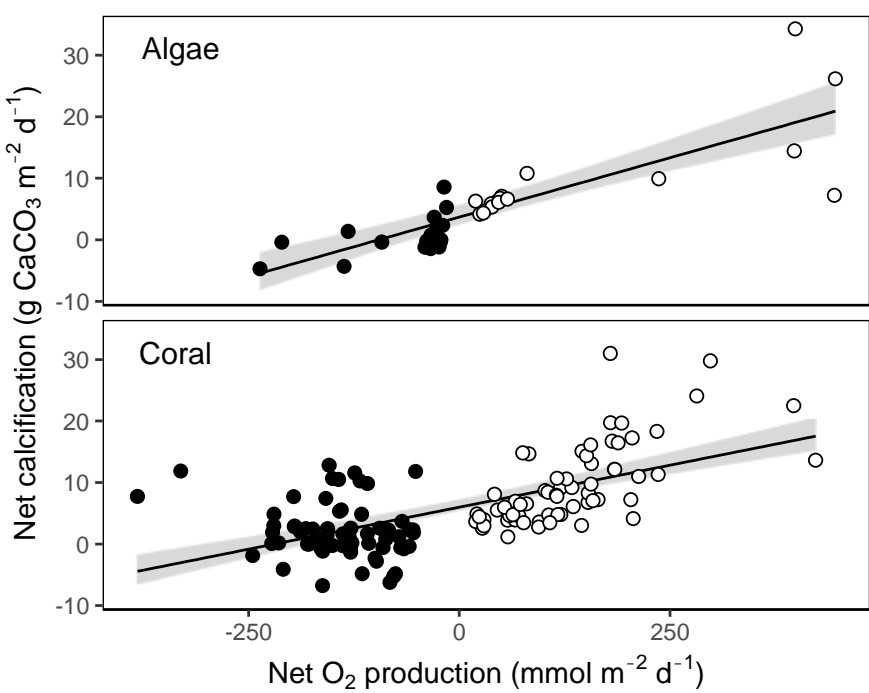


**Figure 7.** Relationship between net calcification rate and net productivity for calcifying algae

(top) and corals (bottom). Open circles indicate rates measured in light and closed circles rates

measured in dark. Linear fits are shown with 95% confidence intervals in gray. For calcified

algae; net calcification = 3.6 + 0.039 × net $O_2$ production (ANOVA: $F_{1,32}$ = 67.0, p <0.001).

For corals; net calcification = 5.99 + 0.027 × net $O_2$ production (ANOVA: $F_{1,126}$ = 82.2, p

<0.001).

655

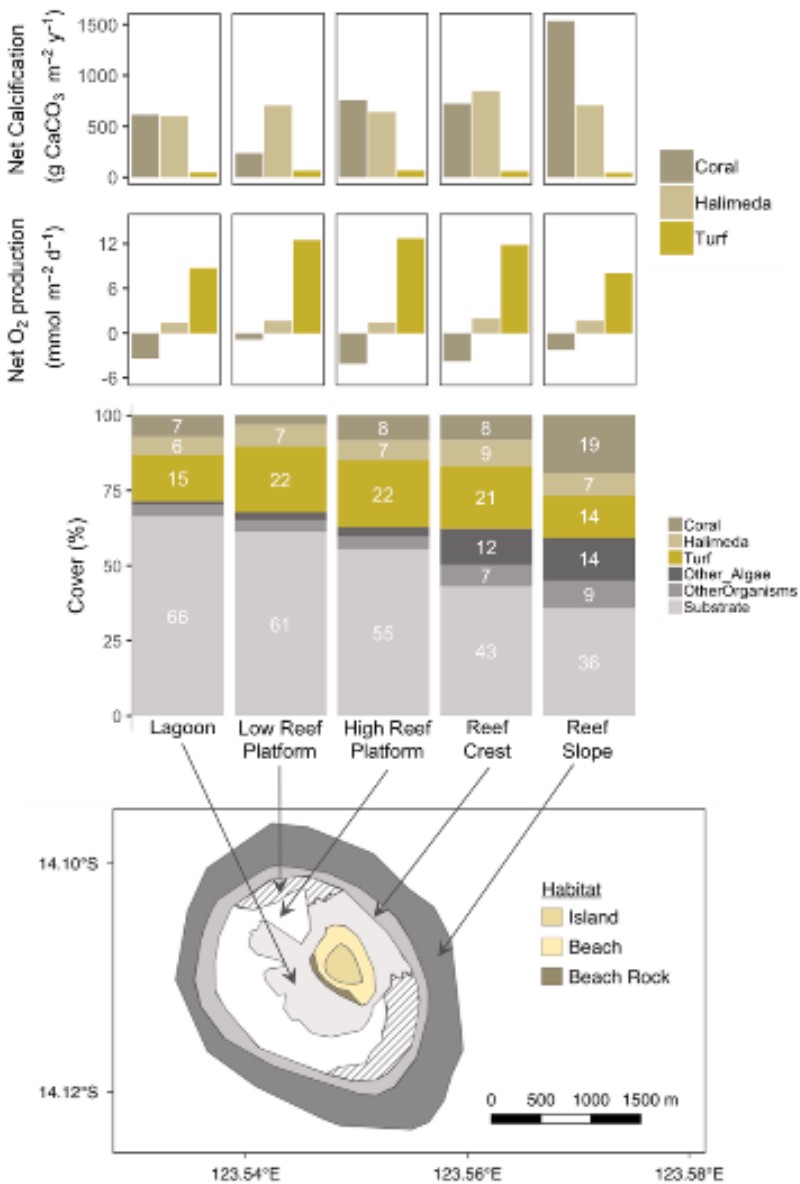

656

**Figure 8.** Map of the reef around Browse Island showing the major habitat types (bottom panel). Reef surface percent cover of coral, *Halimeda*, turf and other categories in each habitat (middle panel) based on drop-camera image analysis data from (Olsen *et al.* 2017). Net calcification and net oxygen production by coral, *Halimeda* and turf per m$^{-2}$ of reef (top two panels) scaled up by multiplying rates obtained from incubations of each taxon by the percent cover in each habitat.

**Tables**
**Table 1.** Taxa measured in on-ship incubation experiments including the number of replicate
specimens measured (one specimen per incubation core). Some of the specimens were not
included in the final analysis due to sampling errors or due to $O_2$ not increasing during both of
the light intervals or not decreasing during both of the dark intervals; the resulting number of
specimens used are shown in brackets.

|       | **Taxa**                | **Apr 2016** | **Oct 2016** | **Oct 2017** |
|-------|-------------------------|--------------|--------------|--------------|
| Algae | *Halimeda opuntia*      | 6 (5)        | 6            | 6            |
|       | Turf algae + substrate  | 6 (5)        | 6            | 6            |
|       | Turf algae              | -            | -            | 6            |
|       | *Sargassum* sp.         | 12           | -            | -            |
|       | *Caulerpa* sp.          | -            | 6            | 6            |
|       | *Galaxaura* sp.         | -            | -            | 6 (5)        |
|       |                         |              |              |              |
| Coral | *Pocillopora* sp.       | 6            | 6            | 6            |
|       | *Goniastrea* sp.        | 6 (5)        | 6            | 6            |
|       | *Porites* sp.           | 5            | 6            | 6            |
|       | *Heliopora* sp.         | -            | 6 (5)        | 6            |
|       | *Acropora* sp.          | -            | 5            | 6            |
|       | *Seriatopora* sp.       | -            | 4            | 6            |
|       |                         |              |              |              |
|       | Seawater control        | -            | -            | 6            |



**Table 2.** Ambient concentrations of parameters measured during incubations (means ± se);
nutrients ($NO_3^-$ + $NO_2^-$ = nitrate + nitrite, $NH_4^+$ = ammonium, $PO_4^{3-}$ = orthophosphate, Si =
silica) and oxygen ($O_2$), total alkalinity (TAlk), Photosynthetically Active Radiation (PAR),
temperature (T) and salinity. Calculated carbonate system parameters (means ± se); $CO_2$
partial pressure ($pCO_2$), concentrations of $HCO_3^-$, $CO_3^{2-}$ and dissolved inorganic carbon
(DIC), and the saturation state of aragonite (Ω Aragonite). In April 2016, two replicate PAR
measurements were taken at 11:00, 12:00 and 13:00 h. In October 2016 and 2017, PAR was
measured every minute and values between 11:00 and 13:00 h averaged.

|  | **Apr 2016** | **Oct 2016** | **Oct 2017** |
|---|---|---|---|
| Number of replicates (n) | 8 | 10 | 12 |
| $NO_3^-$ + $NO_2^-$ (µmol $L^{-1}$) | 0.15 ± 0.04 | 0.05 ± 0.01 | 0.17 ± 0.01 |
| $NH_4^+$ (µmol $L^{-1}$) | 0.12 ± 0.02 | 0.13 ± 0.01 | 0.13 ± 0.01 |
| $PO_4^{3-}$ (µmol $L^{-1}$) | 0.08 ± 0.01 | 0.07 ± 0.00 | 0.09 ± 0.00 |
| Si (µmol $L^{-1}$) | 2.74 ± 0.04 | 2.93 ± 0.04 | 2.30 ± 0.02 |
| $O_2$ (µmol $L^{-1}$) | 19.3 ± 0.19 | 20.8 ± 0.16 | 23.4 ± 0.29 |
| PAR 11–13 h (µE $m^{-2}$ $s^{-1}$) | 1499.6 | 1587.1 | 1587.0 |
| T (°C) | 32.8 ± 0.1 | 31.2 ± 0.1 | 28.3 ± 0.1 |
| Salinity (ppt) | 34.8 | 34.5 | 34.2 |
| TAlk (µmol $L^{-1}$) | NA | 2408 ± 5 | 2390 ± 2 |
| pH | 8.17 ± 0.02 | 8.14 ± 0.02 | 8.11 ± 0.01 |
|  |  |  |  |
| Calculated carbonate system parameters |  |  |  |
| $pCO_2$ (uatm) | NA | 295 ± 14 | 335 ± 17 |
| $HCO_3^-$ (mmol $kg^{-1}$) | NA | 1.61 ± 0.03 | 1.69 ± 0.02 |
| $CO_3^{2-}$ (mmol $kg^{-1}$) | NA | 0.30 ± 0.006 | 0.26 ± 0.006 |
| DIC (mmol $kg^{-1}$) | NA | 1.93 ± 0.02 | 1.97 ± 0.02 |
| Ω Aragonite | NA | 5.02 ± 0.11 | 4.27 ± 0.10 |

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
