# Peer review of "Production and accumulation of reef framework by calcifying corals and macroalgae on"

_EGUsphere, 2022_

## Referee Comment (RC2)

In this paper led by James McLaughlin et al., the authors investigate the rates of CaCO3 production and O2 production/consumption (i.e., photosynthesis and dark Respiration) on different species (green algae, red algae, and corals) to scale up to the community level in a remote island in the Indian Ocean on the west coast of Australia. I greatly appreciated the efforts made by this team to study the functioning of benthic communities, and it is undeniable that much work was done to write this study. However, the writing of the article is still a bit shaky, especially in the references, which are not up-to-date. I do not doubt that this study will be significantly improved after revision and make an excellent reference in benthic functioning.

**Abstract.**

There is a lot of information missing. Especially on the M&M. I mainly see the introduction and the results in this abstract. It would be wise to reduce the results to keep only the key results, add at least two sentences about how you did the experiments (and when), and a sentence in the discussion/conclusion. For example, your M&M part in the abstract stated only: "Specimens of the dominant coral and algal taxa were collected from the reef platform of Browse Island located on the mid-shelf just inside the 200 m isobath off the Kimberley coast. During experimental light/dark incubations, ..." And then, you are already talking about results.
How many species? Light/dark incubations in situ, ex-situ? How many incubation, and how long? What about calcification?

- L22-23: Try to be consistent. One time you do a range of values X - X; another time, you do X to X. stick to one (I prefer the second one)
This comment is valid for the whole manuscript!

**Introduction.**

I had some trouble with the introduction. The ideas are there, but maybe not well arranged. If I followed you correctly, here is your plan: CO2 rising > Functioning and Reef Health > OA and calcification > Algae and calcification > Community metabolism (Calcification and photosynthesis) > Change in pH > Question.
You will see it in my comments below, but I won't just focus on CO2 since you're not looking at it specifically in this study. I will talk about threats more in general. The paragraph on community metabolism and functionning reef health could also be coupled, but I understand your strategy. I might have appreciated a more fluid introduction, but it is still functional. See my comments below.

- L41-43, the way to write it is clumsy. Please rephrase.
I'm a little confused here because you're right to talk about the threats of climate change. But I don't understand why you focus on the OA only when you have the chance to study a remote island: I mean, there is a lower / even zero anthropic pressure (nutrient run-off, sedimentation...). I will expand it in this paragraph.
REF :
1) Hughes, Terry P., et al. "Coral reefs in the Anthropocene." Nature 546.7656 (2017): 82-90.
2) Hughes, Terry P., et al. "Global warming and recurrent mass bleaching of corals." Nature 543.7645 (2017): 373-377.
- L57: Pretty close to what you are doing. I think a similar study to cite would be, especially if you refer to health by looking at metabolism :
1) Carlot, Jeremy, et al. "Scaling up calcification, respiration, and photosynthesis rates of six prominent coral taxa." Ecology and Evolution 12.3 (2022): e8613.
- L60: when you talk about Respiration, please use the standard terminology: "Dark Respiration".
- L65: There are lots of references here. Notably, all the works from Perry et al. (2008, 2012); an older reference would also be Vecsei et al. (2004).
https://scholar.google.com/citations?user=7v8lXzYAAAAJ&hl=en&oi=sra
- L67: If I advocate for the devil, some encrusting corals bring nothing strictly in terms of 3D. I am not sure it is relevant to mention 3D here, especially in this study.
- L69: REF
- L75: A more recent study would be welcome to discuss future projections...
1) Cornwall, Christopher E., et al. "Global declines in coral reef calcium carbonate production under ocean acidification and warming." Proceedings of the National Academy of Sciences 118.21 (2021): e2015265118.
- L80: This statement is critical, given your discussion, and you do not mention the work of Houlbreque et al., in the discussion. Too bad...

- L82: I'm a bit confused about this formulation. Although mass bleaching events are due to thermal stress, bleaching can occur from another cause (i.e., oxic stress due to pollution or disease...). I will remove "during bleaching event" or rephrase it.
- L91: Turf can enhance the productivity of the general system by feeding fish, but not in the way you mean here, in terms of benthic productivity. Am I wrong?
- L95: REF
Remove "like corals" or write "like coral reefs", but it is not necessary; it just bloats the sentence to lose the reader.
- L96, 99: REFs
- L100: What is the ref of the Caribbean?
- L102: It's a little bit fair to extrapolate to the globe with a single reference from 1988 in Tahiti (French Polynesia). Add refs or rephrase
- L123: It's not so difficult to define it, actually. But it becomes very challenging "in situ". Add it.
- L139: This is an excellent paper, but he has updated his results since. It would be nice to see if you are still in the same category.
1) Halpern, Benjamin S., et al. "Recent pace of change in human impact on the world's ocean." Scientific Reports 9.1 (2019): 1-8.

**M&M.**
Overall, it was pretty good. There is some information missing, but I was able to follow everything. I would have appreciated a figure showing the experimental protocol. That would have helped a lot. I'm not sure I understood why there were three dates, maybe I lost the info, or it's not explained. I'm a bit more concerned about the seasonal effect, especially with algae, but I'll come back to that in the discussion. What were the units obtained for Respiration and calcification? Did you do any conversions?

- L158-160: I'm not familiar. Is that a lot for this system? It seems relatively low to me, especially for a remote system
- L174: I'm not sure that estimating the age is relevant. What does it bring to you?
- L185: There is some information missing here. Volume?
- L189: Why 12 for Sargassum and 6 for the others?
- L199: Why 24 cores? I do not understand. Where does 24 come from?
- L210: It's great to have defined the surface, but for the turf, you have mainly defined the surface you extracted. Did you make sure to select the strict minimum? How? There are a lot of limits on this turf component. How did you do in 2017 to have the turf without the substrate (Fig 2)?
- L231: How many replicates triplicates for alkalinity? I am surprised not to see the Dickson et al. (2007) reference in this section.
1) Dickson, Andrew Gilmore, Christopher L. Sabine, and James Robert Christian. Guide to best practices for ocean CO2 measurements. North Pacific Marine Science Organization, 2007.
- L241: Referencing problem with R
- L244: What about the flow? Which pump? What about the light intensity? What about bacterial Respiration?

**Results.**
This part was the weakest. There are some discussion parts in it. See specific comments.

- L273-275: Mix of Discussion / Results, a useless first sentence. Rephrase.
Suggestion: "Nutrient concentrations were low and similar among sampling trips (Table 2)."
- L275-277: Why are you only talking about 2017 values?
- L279-280: Compare what is comparable. First light and then temperature but not light with temperature.
- L281: Which differences?
- L288: Sometimes, a comma would be really useful to understand the flow of your sentence. This is more of a general comment.
- L289-290: Are you talking about bacterial Respiration? If so, please be explicit. This is too bad that you did not apply a correction, even a small one. It would have shown the accuracy of your results.
- L300-302: This part goes into the discussion.
Also, OK, it's a good point, but not necessarily; it can be a response to oxygen stress due to the chamber (according to the EPA, your organisms should not have an oxygen concentration lower than 80%, did you look at it?).
Also, the 9cm diameter for corals is not that small. Especially when you estimated that your corals were up to 7 years old at the beginning!

- L304-306: I think it is better to talk about significance. Indeed, it seems non-significant. I'd prefer to see what's significant or not; mentioning the range of change is great but not enough (you need both). Add this information.
- L315: Once again, minor = non-significant
- L317: The adjusted R2 is not small for ecological models. Would it be better to look only at species? The fact here is that you do not have a lot of samples, and this is why you got lower R2 than your expectations
- L322: That is surprising. Even at night, calcification activity should be less but not negative. Check with other papers for a response to the review.
- L324: Is the ratio Daylight/Night equal to 1 in your location? It would be more accurate to check it instead of assuming it
- L326: Strongly correlated: Prove it (Hint: R2)
- L330: Intertidal, which categories in Fig 7? The high reef platform, I assume. Say it.
Add also error bars. It might be significantly different
- L333: It would be good to recall the cover in Halimeda to get an idea of the contributions
- L338: It would be nice to have a value or a percentage to judge the negligible.

Discussion.
Overall it's good. I wish you would have talked about seasonality, which is totally ignored. It would be nice to compare your values obtained with different methods to support the robustness of your results. Also, it would be interesting to discuss them in the face of climate change since you went in that direction in the introduction. The last paragraph of the discussion before the conclusion is excellent.

- L344: at line 155, only 5m, now more than the double!
- L346: I prefer the word "composed" rather than "inhabited "– the second one being more used in the scientific literature for humans.
- L352-353: Do not get the sentence. Rephrase.
- L369: OK, this is fair, but were results for Halimeda or coral different in other studies?
- L376: I wouldn't necessarily use the terms "Functional groups" since you are always looking at the same functions with different species. I would say species or taxonomic groups.
- L382: Yes and no. They remain autotrophic but get heterotrophic supplies. There is probably a mountain of Houlbreque refs to quote here.
https://scholar.google.com/citations?user=gbjGctUAAAAJ&hl=en&oi=sra
- L386, 414, 422: Only one / add Refs!
- L422: Here, it's good to know what base units you had.
- L463: Excellent paragraph!
- L472: I'm not sure I understood? How do you estimate metabolism aerially? On the other hand, relief is possible with photogrammetric methods.

Figures and Tables.
You have not discussed the nutrients in slide 2. You can refer to the work of
Jacob Allgeier to help you.
https://scholar.google.com/citations?user=1IUg8IwAAAAJ&hl=en&oi=sra

---

## Author Comment (AC1)

**Production and accumulation of reef framework by calcifying corals and macroalgae on a remote Indian Ocean cay.**

**Response to RC1: 'Comment on egusphere-2022-467', Anonymous Referee #1**

General Comments:

This study assessed the metabolic and calcification rates of a variety of cultured, reef-dwelling marine calcifiers and algal taxa found in the anthropogenically-pristine Kimberley bioregion of Western Australia. The values measured were then related to the areal extent of benthic coverage across the various local reef habitats and further argued to provide a baseline for understanding shifts in metabolic and calcification rates in this region in response to environmental stressors (i.e. those that induce bleaching and mortality). While the results presented are surely significant in that they represent novel and important reef metabolic data from a unique location, more time/space could be spent discussing the methods used, assumptions made, and data generated in relation to previously published studies and long outstanding questions in the field (particularly with relation to future impacts of anthropogenic change). The authors should potentially consider reorganizing the key takeaways of the article - particularly in the Discussion section - around these topics as they are currently lacking and/or given short shrift. As it is currently presented, the Discussion section reads as a series of descriptive statements rather than a connected narrative that binds the manuscript together, interprets and provides context for the results of the study, and proposes potential mechanisms and future directions. Overall, I think spending a little more time thinking about the selected location, taxa, observed rates of metabolism/calcification, and trends in O2, pH, and TA in relation to future projected impacts of environmental change in this region (quantitatively, if possible) is a worthwhile endeavor and will only strengthen the impact of this work.

*We thank the reviewer for their insightful and considered comments and suggestions. We have made significant changes to the introduction and discussions sections to address some of the constructive criticisms provided by RC1, which we think adds considerable value and strengthens our manuscript.*

Specific Comments/Questions:

Lines 52 - 61: Just to clarify, it is commonly argued that net community productivity (NCP) rates in many reef ecosystems, while certainly variable over the course of a diel cycle, tend to balance out over longer temporal scales such that nearly all of the organic carbon produced during periods of high photosynthesis is consumed on annual timescales or greater (i.e. Ware et al., 1992; Frankignoulle and Gattuso, 1993; Gattuso et al., 1999; Bates, 2002 and others). Thus, while it is true that $CO_2$ source/sink behavior is possible in coral reefs on short timescales, overall they are believed to be net sources because of high calcification rates.

*We thank the reviewer for providing this clarification. Upon review of the citations provided we have rewritten the second paragraph to make clear any vagueness in our initial draft around rates of photosynthesis and calcification to overall source/sink of $CO_2$ by reefs. We have incorporated the citations of Ware et al, 1992; Gattuso et al, 1993; Gattuso et al, 1995; Smith, 1995; Frankignoulle et al, 1996; Gattuso et al, 1996b and subsequently Kayanne et al, 1995; Gattuso et al, 1996a; Gattuso et al, 1997; Gattuso et al, 1999 to this section of the introduction.*

Lines 123 - 125: I'd argue it's more the magnitude and the net effect of these changes that is difficult to predict rather than conceptually reasoning through the effects themselves. We know the respective impacts of photosynthesis/respiration and calcification/dissolution on many parameters

of carbonate chemistry very well, but "hybrid" organisms that have both NCP and NCC rates (like corals and calcifying algae) make predicting the values of these rates difficult.

*We thank the reviewer for highlighting this. We have sought to address ambiguity by adding "The magnitude of reef contributions to changes in water column chemistry is difficult to predict because of the net effect of local oceanographic conditions, relative abundance of the different members of the reef community and their individual metabolic rates." And further down in the paragraph "The effect on water column chemistry by hybrid organisms like calcifying primary producers such as corals with zooxanthellae and calcifying algae becomes very challenging to measure in situ."*

Lines 131 - 134: Include citations for those that do exist and potentially some discussion on what has been learned and/or what is left to explore or challenge?

*We have restructured the introduction and previous mesocosm experiments is now covered in more detail within the discussion section.*

Section 2.2: What is the rationale/motivation for selecting these particular taxa for incubations - both abundance-wise and otherwise?

*The taxa used for incubations were chosen based on abundance on the reef (i.e., were there enough individuals of a species to collect 6 or more replicates?) and size (< 90 mm) to fit inside our incubation cores without damaging the coral or algal tissue. This is detailed in section 2.2 Algae and coral collection.*

Section 2.3: In the "real" world, PAR has a more parabolic shape with time over the course of a day than the more step-wise shifts induced by the incubation setup. It follows that photosynthesis vs irradiance is often modeled as a hyperbolic tangent function (i.e. Atkinson and Grigg, 1984; Langdon and Atkinson, 2005; Bouman et al., 2018; Bolden et al., 2019). Has any thought been given to what artifacts the simplified 4-hour approach (2 hours light - 2 hours dark) to incubations presented here may have on measured and scaled-up metabolic and calcification rates?

*This is indeed a simplified approach. We have added some wording around this in the discussion.*

Lines 231 - 242: Is this instrument/method calibrated using any kind of standard (such as the Dickson CRMs or another internal standard)?

*The methodology from SOP3b in Dickson et al., 2007 was used to determine the alkalinity for single replicates due to sample volume constraints. The text has been changed to From SOP3b in Dickson et al. 2007, total alkalinity was determined for single replicates to the nearest…*

*Dickson et al. 2007 has been added to the list of references.*

Section 2.6: This is an interesting approach. To clarify, there are four total incubation periods: two, 1-hour light periods and two, 1-hour dark periods. The per-hour rates of O2 (or alkalinity) production or consumption are then multiplied by 24 to get respective light or dark "daily" rates of net productivity (and calcification). However, why is expressing light and dark incubation results in terms of *daily* fluxes valuable? Photosynthesis only occurs during sunlight hours; in your assumptions, there are only 12 hours of photosynthesis and 24 of respiration in a day. Not that these are invalid assumptions, but would it not make more sense to express the light and dark rates on hourly scales *or* do the calculation for the net flux (GPP - R) and express this as 1 daily value for each light+dark pair?

*We have expressed the values as rates per day to enable comparisons with other values in the published literature. The daily values for each light-dark pair per day are also given.*

Lines 280 - 284 (and Table 2): Any comment on why this may be and/or what it implies about seasonal/interannual variability of carbonate chemistry in the offshore waters that are assumed to supply the reef ecosystem?

*Unfortunately, we were not successful in collecting carbonate system parameters in April of 2016 so were unable to compare Austral autumn with Austral spring in October 2016. While we could discuss inter-annual variability, we felt that our sample size was insufficient to confidently elucidate any trends.*

Section 3.2: I think this goes back to my earlier comment - I would think about expressing the respective light and dark fluxes in hourly units rather than daily. I am guessing that the calculations of net autotrophy/heterotrophy are based on light rates minus dark rates. *However* I would double-check to make sure the equations used are consistent with a 12-hour photoperiod and 24-hours of respiration. Including the equations used in the text would be a valuable addition.

*The light flux (and changes in pH) reported are the changes in $O_2$ in the light thereby including both photosynthesis and respiration. The dark flux includes only respiration. The balance between the two is the net flux, which is obviously based on the assumption of equal periods of light and darkness. We acknowledge that the balance between photosynthesis and respiration would not be consistent across all parts of the day. This assumption has now been highlighted more clearly in the discussion.*

Lines 315 - 318: How are the r2 values "adjusted"?

*This is a standard output from an AOV in R. The $R^2$ values are adjusted for the number of parameters in the model relative to the number of points in the design. It thereby takes into account how many independent variables were used to predict the target variable.*

Lines 365 - 369: Why is the alkalinity anomaly technique prone to overestimates of calcification rates? A small clause (and citation?) for this would balance the underestimate assertion for the CaCO3 content/growth method.

*A citation to Hart & Kench, 2007 has been provided in the text to address this.*

Lines 373 - 386: Out of curiosity, to measured pH and alkalinity values produce calculated DIC values (using CO2SYS or seacarb) consistent with the trends in O2 in terms of the magnitude of heterotrophic/autotrophic behavior across taxa?

*Yes, this was pretty consistent with what we expected. Another assessor commented that the apparent dissolution of $CaCO_3$ in some species during night-time was surprising. We have now included some wording around this in the results section.*

Lines 422 - 430: Are there any hypothesized observations/mechanisms for explaining why the calcification rates here are lower than other reported values - particularly as they relate to local open ocean chemistry variability and/or artificats introduced in the incubation + scaling approach?

*We have added commentary around the $CaCO_3$ saturation state for the Northern Indian Ocean and impacts on calcification in the revised discussion.*

Lines 460 - 462: Why were CCA species not included in this incubation study?

*CCA was not readily available in quantities to incubate on the reef flat at low tide when accessibility to collect was available. It was more prominent on the reef slope, but safety concerns prevented us from diving to collect specimens at deeper depths. Bessey et al., 2020 utilised exclosure cages on settlement tiles in the lagoon at Browse Island and no CCA was observed on those after being deployed for 6 months. Turfing algae seemed to dominate the settlement tiles.*

**Technical Comments/Questions:**

Line 69 - No need for possessive. "Coral skeletons are…" is fine.

*Text changed to "Coral skeletons are made from the mineral phase of calcium carbonate…"*

Lines 84 - 85: This concluding sentence reads as a bit of a non sequitur, and this paragraph overall could use some refocusing. I would suggest taking a step back and thinking about the key points the reader is meant to take away from this section. It seems like it's about scleractinian coral contributions to the reef framework and threats to that contribution (based on lines 65-80, but lines 80 - 85 suddenly shift focus to metabolism.

*We have re-written the introduction and through that process the text in this paragraph has been changed significantly or removed.*

Line 87 - I think this first sentence could be stated more concisely. "Reef algae are also an important structural component of coral reef ecosystems. Their morphological diversity provides…"

*Text change to "Reef algae are also an important structural component of coral reef ecosystems."*

Line 95 - Sentence could be more concise. "Calcified macroalgae can also contribute significantly to the deposition of carbonates in coral reef environments."

*Text changed to "Calcified macroalgae can also contribute significantly to the deposition of carbonates in coral reef environments."*

Line 99 - "make it a major contributor".

*Text changed to "Production rates of Halimeda make it major contributor to CaCO3 in reefs in…"*

Lines 99 - 102: Here and throughout, be careful and consistent with the use of the term "production" to refer to organic carbon/oxygen production vs CaCO3 precipitation.

*We have changed the text in the introduction so it now reads "Calcification rates of Halimeda make it a major contributor to CaCO3 in reefs in the Caribbean…"*

Line 277 (and elsewhere): Consider expressing O2 concentration in molar units (as you do in subsequent discussions). It would be more consistent with the other measured chemical constituents and allow readers to think about potential stoichiometric relationships between variables more easily.

*We have changed the units for oxygen from mg $L^{-1}$ to umol $L^{-1}$ for consistency.*

Lines 290 - 292: This sentence could be more concise and clear (I think). "In light incubations, O2 productivity fluxes were postitive across all taxa."

*Text changed to "In the light incubations O2 productivity fluxes were positive for all taxa (Fig. 4, top panel)."*

Lines 395 - 398: This is repeated at line 364.

*The repeated line has been removed.*

---

## Author Comment (AC2)

**Production and accumulation of reef framework by calcifying corals and macroalgae on a remote Indian Ocean Cay.**

**Response to RC2: 'Comment on egusphere-2022-467', Anonymous Referee #2**

In this paper led by James McLaughlin et al., the authors investigate the rates of CaCO3 production and O2 production/consumption (i.e., photosynthesis and dark Respiration) on different species (green algae, red algae, and corals) to scale up to the community level in a remote island in the Indian Ocean on the west coast of Australia. I greatly appreciated the efforts made by this team to study the functioning of benthic communities, and it is undeniable that much work was done to write this study. However, the writing of the article is still a bit shaky, especially in the references, which are not up-to-date. I do not doubt that this study will be significantly improved after revision and make an excellent reference in benthic functioning.

*We thank RC2 for the time spent providing constructive criticism and comments on our manuscript. We greatly appreciate the reviewers suggested additional reference material for us to consider implementing into this manuscript. We feel in addressing the comments below, the effort has added considerable value to our work resulting in a much more impactful summation for our study of the reef at Browse Island.*

**Abstract.**

There is a lot of information missing. Especially on the M&M. I mainly see the introduction and the results in this abstract. It would be wise to reduce the results to keep only the key results, add at least two sentences about how you did the experiments (and when), and a sentence in the discussion/conclusion. For example, your M&M part in the abstract stated only: "Specimens of the dominant coral and algal taxa were collected from the reef platform of Browse Island located on the mid-shelf just inside the 200 m isobath off the Kimberley coast. During experimental light/dark incubations, ..." And then, you are already talking about results. How many species? Light/dark incubations in situ, ex-situ? How many incubations, and how long? What about calcification?

*We have restructured the abstract so that it now includes more information about the methods we used.*

- L22-23: Try to be consistent. One time you do a range of values X - X; another time, you do X to X. stick to one (I prefer the second one)

*We have changed the text in the abstract so it now reads "all algae were net autotrophic producing 6 to 111 mmol $O_2$ $m^{-2}$ $day^{-1}$."*

This comment is valid for the whole manuscript!

*We have changed the values ranges within the text of the manuscript from "X – X" to "X to X" throughout to ensure consistency.*

**Introduction.**

I had some trouble with the introduction. The ideas are there, but maybe not well arranged. If I followed you correctly, here is your plan: CO2 rising > Functioning and Reef Health > OA and calcification > Algae and calcification > Community metabolism (Calcification and photosynthesis) >

Change in pH > Question. You will see it in my comments below, but I won't just focus on CO2 since you're not looking at it specifically in this study. I will talk about threats more in general. The paragraph on community metabolism and functioning reef health could also be coupled, but I understand your strategy. I might have appreciated a more fluid introduction, but it is still functional. See my comments below.

*We thank the reviewer for the constructive criticisms of our introduction and especially for supplying a number of more recent references. We have re-written the introduction to emphasize the calcification rates, productivity and reef functioning more in the context of general pressures that can act to constrain these processes. As the reviewer so rightly points out our study is not specific to ocean acidification, and we have adjusted accordingly.*

- L41-43, the way to write it is clumsy. Please rephrase.

I'm a little confused here because you're right to talk about the threats of climate change. But I don't understand why you focus on the OA only when you have the chance to study a remote island: I mean, there is a lower / even zero anthropic pressure (nutrient run-off, sedimentation...). I will expand it in this paragraph.

*We have re-written the introduction to emphasize the calcification rates, productivity and reef functioning as the focus instead of ocean acidification.*

REF :

1) Hughes, Terry P., et al. "Coral reefs in the Anthropocene." Nature 546.7656 (2017): 82-90.

2) Hughes, Terry P., et al. "Global warming and recurrent mass bleaching of corals." Nature 543.7645 (2017): 373-377.

- L57: Pretty close to what you are doing. I think a similar study to cite would be, especially if you refer to health by looking at metabolism:

1) Carlot, Jeremy, et al. "Scaling up calcification, respiration, and photosynthesis rates of six prominent coral taxa." Ecology and Evolution 12.3 (2022): e8613.

*We have moved this text in the re-written introduction to the last paragraph with updated citations and it now reads "Metabolic measurements of reef organisms are necessary to characterize reef health in terms of fundamental processes such as photosynthesis, respiration and calcification (Madin et al., 2016; Carlot et al., 2022)."*

- L60: when you talk about Respiration, please use the standard terminology: "Dark Respiration".

*We have changed the text to now read "i.e., the difference between gross primary production and dark respiration,…"*

*We have changed "respiration" to "dark respiration" where suitable throughout.*

- L65: There are lots of references here. Notably, all the works from Perry et al. (2008, 2012); an older reference would also be Vecsei et al. (2004).

*We have included references to Vecsei, 2004; Perry et al., 2008; and Perry et al., 2012 to the corresponding text.*

https://scholar.google.com/citations?user=7v8lXzYAAAAJ&hl=en&oi=sra

- L67: If I advocate for the devil, some encrusting corals bring nothing strictly in terms of 3D. I am not sure it is relevant to mention 3D here, especially in this study.

*Reference to 3 dimension of reef structure has been removed. The text now reads "Scleractinian corals are primary reef builders in tropical environments producing CaCO3 through skeletal deposition."*

- L69: REF

*We have added reference to Kleypas and Yates, 2009.*

- L75: A more recent study would be welcome to discuss future projections...

1) Cornwall, Christopher E., et al. "Global declines in coral reef calcium carbonate production under ocean acidification and warming." Proceedings of the National Academy of Sciences118.21 (2021): e2015265118.

*We have added reference to Cornwall et al., 2021. The text now reads "… with the majority of coral reefs unable to maintain positive net carbonate production globally by 2100 (Cornwall et al., 2021)."*

- L80: This statement is critical, given your discussion, and you do not mention the work of Houlbreque et al., in the discussion. Too bad...

*Reference to relevant works by Houlbreque et al. has been added to parts of the discussion.*

- L82: I'm a bit confused about this formulation. Although mass bleaching events are due to thermal stress, bleaching can occur from another cause (i.e., oxic stress due to pollution or disease...). I will remove "during bleaching event" or rephrase it.

*Reference to bleaching events has been removed. The text now reads "Rates of primary production and dark respiration increase but community excess organic production decreases dramatically in reefs under thermal stress (Kayanne et al., 2005)."*

- L91: Turf can enhance the productivity of the general system by feeding fish, but not in the way you mean here, in terms of benthic productivity. Am I wrong?

*You are correct. We've changed the text so it now reads* "… but inconspicuous turfs and encrusting coralline algae contribute substantially to reef benthic primary resources in these areas…" *to clear up the ambiguity around the term "primary production" which generally means the sequestration of carbon.*

- L95: REF

*We have added reference to Purcell and Bellwood, 2001*

Remove "like corals" or write "like coral reefs", but it is not necessary; it just bloats the sentence to lose the reader.

*We have changed the text so it now reads "Calcified macroalgae can also contribute significantly to the deposition of carbonates."*

- L96, 99: REFs

*We have added reference to Vroom et al., 2003, Smith et al., 2004, Nelson, 2009.*

- L100: What is the ref of the Caribbean?

*We have added reference to Blair and Norris, 1988, and Nelson, 2009.*

- L102: It's a little bit fair to extrapolate to the globe with a single reference from 1988 in Tahiti (French Polynesia).

*We've changed the text to now read "In certain locations precipitation of Calcium carbonate can approach 2.9 kg CaCO3 m−2 yr−1, positioning Halimeda as a major contributor to carbonate budgets within shallow waters around the globe." With a reference to Price et al., 2011 which contains estimates of Halimeda contribution to reef calcification from around the globe.*

- L123: It's not so difficult to define it, actually. But it becomes very challenging "in situ". Add it.

*We have changed the text so it now reads "How calcifying primary producers such as corals with zooxanthellae and calcifying algae affect water column chemistry becomes very challenging to measure in situ."*

- L139: This is an excellent paper, but he has updated his results since. It would be nice to see if you are still in the same category.

1) Halpern, Benjamin S., et al. "Recent pace of change in human impact on the world's ocean." Scientific Reports 9.1 (2019): 1-8.

*We have included reference to Halpern et al., 2019 in addition to his 2008 paper.*

**M&M.**

Overall, it was pretty good. There is some information missing, but I was able to follow everything. I would have appreciated a figure showing the experimental protocol. That would have helped a lot. I'm not sure I understood why there were three dates, maybe I lost the info, or it's not explained. I'm a bit more concerned about the seasonal effect, especially with algae, but I'll come back to that in the discussion.

*Photographs of experimental setup added as Figure 2, all subsequent figure captions changed to reflect this addition.*

What were the units obtained for Respiration and calcification? Did you do any conversions?

*Respiration rates were calculated as the $O_2$ produced (measured in micromol per L per unit time and converted in mmol per day per incubation core) divided by the surface are of the algal or coral specimen (in $m^2$) for units in mmol per day per $m^2$.*

*Calcification rates were calculated as the $CaCO_3$ produced (in g per day) per surface are of the algal or coral specimen (in $m^2$). These units are converted from pH and alkalinity using the 'seacarb' package in R as stated*

- L158-160: I'm not familiar. Is that a lot for this system? It seems relatively low to me, especially for a remote system

*This is quite low, but not atypical for the region.*

- L174: I'm not sure that estimating the age is relevant. What does it bring to you?

*The age is important as a point of reference for the reader as calcification and respiration rates are likely to differ between juvenile and adult specimens.*

- L185: There is some information missing here. Volume?

*Text moved to L202. Sampling volume is stated in the preceding sentence at L200.*

- L189: Why 12 for Sargassum and 6 for the others?

*Text changed to address ambiguity and now reads "Depending upon abundance, individual specimens of algae and coral were placed in 6– to 12 replicate incubation cores per taxa except where not enough individuals could be found."*

*Essentially, Sargassum was found in greater quantities in April 2016 than other taxa so that is why we incubated 12 plants as opposed to 6.*

- L199: Why 24 cores? I do not understand. Where does 24 come from?

*6 cores fit evenly spaced in each of the 60 L tubs where self-shading is minimised. We used 24 cores because that is the total number we could fit into the 4 tubs.*

*Text at L184 changed to "Six 1.56 L clear Perspex incubation cores (24 total per incubation) fitted with stirring caps, were placed in each holding tank and spaced evenly apart to minimise shading (Fig. 2).*

- L210: It's great to have defined the surface, but for the turf, you have mainly defined the surface you extracted.

Did you make sure to select the strict minimum? How? There are a lot of limits on this turf component. How did you do in 2017 to have the turf without the substrate (Fig 2)?

*Turf without a substrate were flat two-dimensional pieces of turf (not attached to pieces of rock), hence we were able to measure the surface from a simple photograph. For turf with substrate, the pieces were three-dimensional, hence why we had to use a wax dipping method to estimate the surface area. The turf 'fronds' are so small that there is no way to measure the actual combined surface area of them.*

- L231: How many replicates triplicates for alkalinity? I am surprised not to see the Dickson et al. (2007) reference in this section.

*The methodology from SOP3b in Dickson et al., 2007 was used to determine the alkalinity for single replicates due to sample volume constraints. The text has been changed to From SOP3b in Dickson et al. 2007, total alkalinity was determined for single replicates to the nearest…*

*Dickson et al. 2007 has been added to the list of references.*

- L241: Referencing problem with R

*Citation corrected. Now reads (R Core Team, 2018).*

- L244: What about the flow? Which pump? What about the light intensity? What about bacterial Respiration?

*Relevant information on the pump used is now provided at L179 "… filled with seawater and connected to a flow-through seawater system driven by an Ozito PSDW-350 watt Dirty Water Submersible Water Pump with a maximum flow rate of 7,000 litres/hour…"*

*L183 includes information on instrumentation used to measure light intensity "The intensity of photosynthetically active radiation (PAR) was recorded for each set of incubations with a HOBO Micro Station logger (H21-002, Onset) placed inside one of the tanks."*

*Bacterial respiration was not specifically determined for this set of experiments however in the seawater controls $O_2$ changed by < 0.01 mmol $h^{-1}$ in both light and dark incubations, showing that the contribution of any organisms (i.e. bacteria, phytoplankton, etc.) in the seawater itself to either light driven productivity or dark respiration was minimal.*

**Results.**

This part was the weakest. There are some discussion parts in it. See specific comments.

- L273-275: Mix of Discussion / Results, a useless first sentence. Rephrase.

Suggestion: "Nutrient concentrations were low and similar among sampling trips (Table 2)."

*Text changed to "Nutrient concentrations were low and similar among sampling trips (Table 2), as is characteristic of tropical Eastern Indian Ocean offshore waters (McLaughlin et al., 2019)."*

- L275-277: Why are you only talking about 2017 values?

*We have changed the text so it now presents the lower and upper ranges of the nutrient concentrations measured.*

- L279-280: Compare what is comparable. First light and then temperature but not light with temperature.

*Sentence split to alleviate ambiguity of comparisons and now reads as "PAR levels were 1500 to – 1587 µE m−2 s−1 and slightly higher in October. Temperatures were 28.3 to 32.8℃ and highest in April."*

- L281: Which differences?

*Text has been changed and now reads "Carbonate system parameters were not obtained for April 2016, and some minor differences in pCO2, $HCO3^-$, $CO3_2^-$, DIC and Ω Aragonite were noted between October 2016 and 2017 (Table 2)." to highlight the parameters measured in 2016 and 2017.*

- L288: Sometimes, a comma would be really useful to understand the flow of your sentence. This is more of a general comment.

*Text changed and now reads "Changes in dissolved O2 differed among taxa, and between light and dark incubations." Comment about commas understood.*

- L289-290: Are you talking about bacterial Respiration? If so, please be explicit. This is too bad that you did not apply a correction, even a small one. It would have shown the accuracy of your results.

*Yes, bacterial and plankton respiration. As stated in the text, the controls displayed values that were so small compared to the production and consumption of oxygen in the algal and coral incubations that no correction was deemed necessary.*

- L300-302: This part goes into the discussion.

*We have moved this text into the discussion section.*

Also, OK, it's a good point, but not necessarily; it can be a response to oxygen stress due to the chamber (according to the EPA, your organisms should not have an oxygen concentration lower than 80%, did you look at it?).

*We thank the reviewer for highlighting this. We were unaware of this threshold and will consider investigating in future studies.*

Also, the 9cm diameter for corals is not that small. Especially when you estimated that your corals were up to 7 years old at the beginning!

*We appreciate the reviewer clarifying size age relationships for coral colonies and have amended the text.*

- L304-306: I think it is better to talk about significance. Indeed, it seems non-significant. I'd prefer to see what's significant or not; mentioning the range of change is great but not enough (you need both). Add this information.

*We are not sure what the reviewer is referring to here as being non-significant? A change in pH of 0.03-0.25 is large. The choice of word (minor) is deliberate so as not to confuse this with statistical significance.*

- L315: Once again, minor = non-significant

*The choice of word (minor) is deliberate so as not to confuse this with statistical significance.*

- L317: The adjusted R2 is not small for ecological models. Would it be better to look only at species? The fact here is that you do not have a lot of samples, and this is why you got lower R2 than your expectations

*It was not really possible or practical to look at individual species due to low sample numbers. This analysis supports the observation that changes in pH were driven by metabolic uptake and release of $CO_2$ even with the low $r^2$ values, which can be explained by the fact that there is variability among individual specimens and species.*

- L322: That is surprising. Even at night, calcification activity should be less but not negative. Check with other papers for a response to the review.

*We agree that this was surprising and may be an artefact of the experimental conditions. This has been added to the results section and the language of what caused the change has been toned down.*

- L324: Is the ratio Daylight/Night equal to 1 in your location? It would be more accurate to check it instead of assuming it

*Text changed to "The resulting net calcification rates (based on equal periods of light and dark - monthly average sunrise and sunset at Browse Island of 0552 and 1739 for April, and 0519 and 1754 for October; WillyWeather, 2022)"*

*WillyWeather, 2022 website citation has been added to the reference list.*

- L326: Strongly correlated: Prove it (Hint: R2)

*We have provided the results of an ANOVA with a p-value in the figure legend which we believe provides a more robust analysis.*

- L330: Intertidal, which categories in Fig 7? The high reef platform, I assume. Say it.

*Text changed to "In intertidal habitats (lagoon and high reef platform) around Browse Island, the estimated relative contributions…"*

Add also error bars. It might be significantly different

*Figure 7 represents a conceptual diagram of the reef metabolic processes, hence, error bars are not included.*

- L333: It would be good to recall the cover in Halimeda to get an idea of the contributions

*Text changed to include % cover now reads "In intertidal habitats around Browse Island, the estimated relative contributions of coral (8% cover) and Halimeda (7% cover)…"*

- L338: It would be nice to have a value or a percentage to judge the negligible.

*Text now reads "Galaxaura, which had high measured rates of productivity and calcification, was extremely rare (0.02 % total cover found only in October 2017; Olsen et al., 2017) and thus its contribution to community calcification and productivity were negligible."*

Discussion.

Overall it's good. I wish you would have talked about seasonality, which is totally ignored. It would be nice to compare your values obtained with different methods to support the robustness of your results. Also, it would be interesting to discuss them in the face of climate change since you went in that direction in the introduction. The last paragraph of the discussion before the conclusion is excellent.

*Unfortunately, we were not successful in collecting carbonate system parameters in April of 2016 so were unable to compare Austral autumn with Austral spring in October 2016. While we could discuss inter-annual variability, we felt that our sample size was insufficient to confidently elucidate any trends.*

- L344: at line 155, only 5m, now more than the double!

*Text clarified at L355. Now reads "Browse Island has the only emergent mid-shelf reef in the Kimberley bioregion with semidiurnal tides reaching a maximum range of < 5 m, half of those tides experienced by reefs closer to the coast."*

- L346: I prefer the word "composed" rather than "inhabited "– the second one being more used in the scientific literature for humans.

*Text changed and now reads "The region contains thousands of islands with a total reef area estimated to be ~2000 km$^2$ (Kordi and O'Leary, 2016), composed of inhabited by diverse coral reef and seagrass communities…"*

- L352-353: Do not get the sentence. Rephrase.

*The text has been changed to "The region-wide averages of coral cover and macroalgal cover are 23.8% and 7.1% (Richards et al., 2018) respectively. However, the relationship at Browse Island is reversed with macroalgae more dominant at 28% total cover to that of coral at 9% total cover."*

- L369: OK, this is fair, but were results for Halimeda or coral different in other studies?

*We've added an additional reference to Houlbreque et al., 2009.*

- L376: I wouldn't necessarily use the terms "Functional groups" since you are always looking at the same functions with different species. I would say species or taxonomic groups.

*Text changed and now reads"… but the metabolic measurements showed clear differences among taxonomic groups."*

- L382: Yes and no. They remain autotrophic but get heterotrophic supplies. There is probably a mountain of Houlbreque refs to quote here.

*Text changed to "Although autotrophic, our data indicates that the majority of the corals we studied utilise heterotrophic supply through feeding to help sustain growth in addition to photosynthesis by zooxanthellae."*

*We have added some relevant references to work done by Houlbreque et al. in parts of the discussion.*

https://scholar.google.com/citations?user=gbjGctUAAAAJ&hl=en&oi=sra

- L386, 414, 422: Only one / add Refs!

- L422: Here, it's good to know what base units you had.

*As written in the following text we used g $m^{-2}$ $year^{-1}$*

- L463: Excellent paragraph!

- L472: I'm not sure I understood? How do you estimate metabolism aerially? On the other hand, relief is possible with photogrammetric methods.

*We used photogrammetric quadrats collected on the trips to estimate the cover of coral and macroalgal species. From this cover data we scaled up our metabolic rates to estimate the community metabolism for the reef habitats around the island. We've clarified this in the redrafted discussion of the paper.*

**Figures and Tables.**

*Photographs of experimental setup added as Figure 2, all subsequent figure captions changed to reflect this addition.*

You have not discussed the nutrients in slide 2. You can refer to the work of Jacob Allgeier to help you.

https://scholar.google.com/citations?user=1IUg8IwAAAAJ&hl=en&oi=sra

*We have added a paragraph in the discussion section to address the nutrient data found in Table 2 and provide some commentary and wider context for our results.*

---

## Author Response (AR2)

**Anonymous referee #1 Comments**

Suggestions for revision or reasons for rejection (will be published if the paper is accepted for final publication)

*I highly suggest the authors consider re-outlining the Introduction as it will help focus the details of this key section of the manuscript and further highlight the significance of their work. Many of the paragraphs in this section currently read as though they are each the "first" paragraph in an Introduction, rather than a series of connected ideas. I wholeheartedly agree with the authors that having baseline metabolic and benthic community composition data from a relatively pristine coral reef ecosystem is important, but the "why" in relation to compounding environmental/climate stressors and/or the need to compare to ecosystems with different benthic community composition is somewhat lost when interpreting through these seemingly disjointed paragraphs and often repeated but vague statements about biogeochemical cycling and metabolism within reef ecosystems. That said, clarity and details in the Methods, Experimental Results, and Discussion sections are much improved and provide an appropriate level context for the study.*

*Line Edits/Comments/Technical Corrections*

*Summary of Intro P1: Stressors causing changes in reef ecosystems?*

*Summary of Intro P2: Ecosystem services provided by modern reefs?*

*Summary of Intro P3: Capturing metabolism in coral reef ecosystems?*

*Summary of Intro P4: Calcification on reefs and the threat of OA?*

*Summary of Intro P5: Back to metabolism on reefs?*

*Summary of Intro P6: The important roles of benthic algae in reef ecosystems?*

*Summary of Intro P7: Influence of reef metabolism on seawater chemistry?*

*Summary of Intro P8: Uniqueness of the opportunity to study the Kimberley Bioregion of NW Australia?*

**We appreciate the reviewer's feedback regarding our MS and appreciate their time in providing comment. As suggested, we have re-outlined the introduction (see below). We feel this has resulted in more focus and better-connected ideas in the re-written introductory section.**

**Summary of Intro P1 - Ecosystem services provided by modern reefs/Stressors causing changes in reef ecosystems**

**Summary of Intro P2 - Calcification on reefs and the threat of OA**

**Summary of Intro P3 – Reef metabolism**

**Summary of Intro P4 – Algal contributions to reef metabolism and calcification**

**Summary of Intro P5 - Uniqueness of the opportunity to study the Kimberley Bioregion of NW Australia and study outline/goals**

*Line 41: "current" rather than "currently"?*

**We have made the suggested edit.**

*Line 43: Citation for the impacts of "recurrent large scale weather events"?*

**We have included a citation and reference to Moore et al., 2012 to address large scale weather events such as marine heat waves.**

*Lines 41 - 47: A lot packed into this sentence; consider breaking up and refocusing. Also, a bit unclear on the link between eutrophication, sedimentation, and increasing SST and OA. I think breaking this sentence up could help with this though.*

**We have spliced and restructured the sentence and it now reads "With the current unprecedented rate of environmental change, coral reefs face growing pressures. These include localised eutrophication (Hewitt et al., 2016) and sedimentation (Hughes et al, 2017a), to larger scale recurrent weather events (marine heat waves, etc; Moore et al., 2012), and rising atmospheric greenhouse gases (especially carbon dioxide, $CO_2$; IPCC, 2014) resulting in increasing ocean temperatures (due to atmospheric heat absorption) and ocean acidification (OA) (Hoegh-Guldberg, 2007; Doney et al., 2009; Perry et al., 2018)."**

*Lines 52 - 57: Shouldn't these sentences come before ecosystem services are introduced in the last sentence of the preceding paragraph?*

**We have moved lines 52 – 57 up into the preceding paragraph as suggested.**

*Lines 64 - 66: The way this sentence is written is counter to lines 71 - 75.*

**We have re-written the introduction to address these comments.**

Lines 71 - 75: Adding the timescales (i.e. seasonal, annual, etc.) over which reef flats are generalized to be net sources of $CO_2$ to the atmosphere may help reconcile this statement with the "high" calcification and production rates mentioned above.

**We have re-written the introduction to address these comments.**

*Line 306: Rogue comma.*

**We have removed the comma.**

*Line 309: Space needed between 34.8 and ppt.*

**We have added a space before and after 34.8 ppt.**

*Lines 313 - 314: Check subscripts and superscripts.*

**We have adjusted the sub- and super-scripts in the sentence.**

*Line 355: Rogue underscore?*

**We have removed the superfluous underscore.**

*Lines 378 - 396: Not sure if this paragraph is well-positioned here. A lot of the statements in this paragraph read as site descriptions that may serve a stronger role in the Methods section when introducing Browse Island. Additionally, the "general" discussion of the influences of production and calcification on ambient seawater chemistry has already been brought up in the Introduction.*

**We have moved some of the relevant text into the conclusions as it seemed a better fit there. The text already covered in the introduction has been removed.**

*Lines 398 - 417: I think this is a much stronger starting paragraph for the Discussion section.*

**We appreciate this feedback.**

*Lines 413 - 417: Run-on sentence. Consider a full stop after "elsewhere" and starting a new sentence with "For example…"*

**We have divided the sentence in to two parts as suggested.**

*Line 415: Rogue space after O2.*

**We have removed the superfluous space after O2.**

*Lines 448 - 453 (and throughout): Be mindful of run-on sentences.*

**We have divided the sentence in to two parts as suggested.**

*Lines 453 - 455: "… or use of radioisotopes \*was\* limited."*

**The sentence has been changed to "…or use of radioisotopes was limited."**

*Lines 584 - 586: Repeated in 586 - 587.*

**We have removed the repeated sentence from the text.**

*Figures 3, 4, and 6: Axes are unreadable/corrupted.*

**The text in the axes being difficult to read was a function of the size to the figures. We have increased the size slightly and this seems to have alleviated the letters being squished together resulting in text that is more legible.**